# OmniPlay: Benchmarking Omni-Modal Models on Omni-Modal Game Playing

## Abstract

While generalist foundation models like Gemini and GPT-4o demonstrate impressive multi-modal competence, existing evaluations fail to test their intelligence in dynamic, interactive worlds. Static benchmarks usually lack agency, while interactive benchmarks typically ignore crucial auditory and temporal cues. To bridge this evaluation chasm, we introduce OmniPlay, a diagnostic benchmark designed not just to evaluate, but to probe the fusion and reasoning capabilities of agentic models across the primary audiovisual and textual modalities. Built on a core philosophy of modality interplay, OmniPlay acts as a "wind tunnel" for AI, comprising a suite of five game environments that systematically create scenarios of both complementarity and conflict. Our comprehensive evaluation of six leading omni-modal models reveals a critical dichotomy: they exhibit superhuman performance on high-fidelity memory tasks but suffer from systemic failures in challenges requiring robust reasoning and strategic planning. Through targeted diagnostic experiments, including modality conflict and In-Context Learning (ICL), we demonstrate that this fragility stems from brittle fusion mechanisms rather than a simple lack of task adaptation. We further uncover a counter-intuitive "less is more" phenomenon, where removing sensory information can paradoxically improve performance. Our findings suggest that the path toward robust AGI requires a research focus beyond scaling to explicitly address synergistic fusion. Our platform is available for anonymous review at `https://anonymous.4open.science/r/omniplay`.

## 1 Introduction

Generalist foundation models such as Google's Gemini (Team et al., 2023) and OpenAI's GPT-4o (OpenAI, 2024) have recently accelerated progress toward Artificial General Intelligence (AGI). These models process text, image, audio, and video with impressive competence. However, a model's intelligence is best measured not only by its passive perception of static data, but also by its ability to reason and act through interactive decision-making in a dynamic, sensorially rich world. This raises a critical question: how can we effectively evaluate a model's ability to integrate multi-modal understanding with real-world interaction?

We argue that the existing evaluation landscape is split by a fundamental gap. On one side, many prominent multimodal benchmarks remain static, testing passive understanding in formats like VQA (Antol et al., 2015), MMBench (Liu et al., 2024), and SEED-Bench (Li et al., 2023). These lack crucial dimensions of agency and long-term planning. On the other side, a second wave of benchmarks has shifted toward interactive environments such as ALFWorld (Shridhar et al., 2021) and WebArena (Zhou et al., 2023). While this move towards agency is vital, the majority of these interactive agents operate with limited modalities, typically confined to vision-language inputs, which restricts their ability to process auditory or complex temporal cues.

This paper argues that integrating a broad spectrum of modalities is a foundational requirement for robust omni-modal agency: an agent's capacity to perceive, reason, and make decisions by fluidly integrating inputs across senses. The core challenge lies in managing the complexities of modality interplay. On one hand, sensory inputs can be complementary, where one modality compensates for the limitations of another — for instance, using audio cues to navigate when vision is occluded. Leveraging this complementarity is crucial for effective decision-making. On the other hand, in-

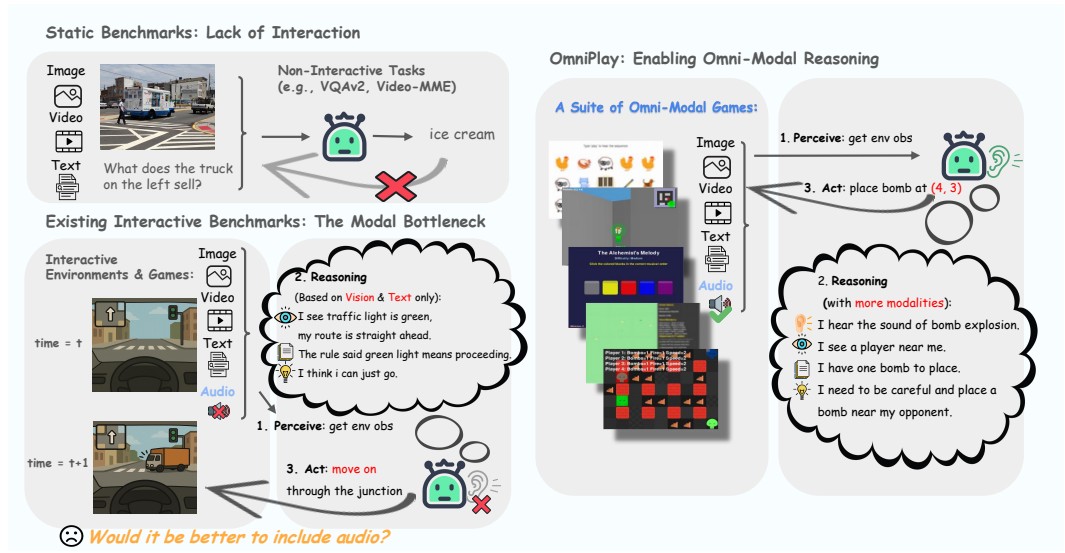

Figure 1: An illustration of the core motivation for OmniPlay. **Left:** Prior benchmarks suffer from two key limitations. Static benchmarks (e.g., VQA) lack interaction and agency. Existing interactive benchmarks are often limited to vision and text, ignoring critical modalities like audio. **Right:** OmniPlay addresses these gaps by providing an interactive, omni-modal environment. It enables agents to perform synergistic reasoning by integrating information across primary modalities (e.g., combining visual, auditory, and textual cues) for more robust decision-making.

puts across modalities can be conflicting. For instance, receiving contradictory visual and auditory commands will create ambiguity that degrades performance. This critical capability to synergize complementary information and resolve sensory conflicts remains largely underexplored by current methodologies.

To diagnose these foundational weaknesses, we introduce **OmniPlay**, a benchmark designed not just to evaluate, but to diagnose the omni-modal fusion and reasoning capabilities of agentic models, as illustrated in Figure 1. OmniPlay is built upon a core philosophy of modality interplay. Across a suite of five distinct games, we develop scenarios that require the synergistic fusion of varying modality combinations (e.g., image-audio-text, video-audio). By systematically manipulating modality complementarity and conflict, OmniPlay functions as a diagnostic toolkit to answer critical questions: Can the model resolve contradictory inputs, or does it fail silently? Is the observed brittleness a fundamental incapacity or merely a lack of task adaptation?

Our primary contributions are:

1. We introduce OmniPlay, the first interactive benchmark designed to diagnose an agent's synergistic fusion, conflict resolution, and adaptive reasoning under controlled modality interplay across audiovisual and textual modalities.

2. We design a suite of five games built on the principle of modality interplay. We empirically validate the benchmark's design through modality ablation, difficulty calibration, and human baseline comparisons.

3. Our comprehensive analysis reveals a critical finding: while models can adapt to strategic tasks via In-Context Learning, they remain fundamentally brittle under modality conflict. We further demonstrate systemic weaknesses via the "less is more" paradox.

4. We commit to open-sourcing the entire OmniPlay platform, including all environments, baseline agents, and evaluation protocols under an MIT license, to facilitate relevant research.

## 2 RELATED WORK

Our research is positioned at the intersection of multimodal evaluation and interactive agent learning. We structure our review by first discussing static benchmarks to highlight the need for agency, then examining interactive benchmarks to reveal their modal bottleneck, and finally arguing that the rise of omni-modal models has turned this bottleneck into a critical evaluation chasm that OMNIPLAY aims to bridge.

### 2.1 STATIC MULTIMODAL BENCHMARKS: PERCEPTION WITHOUT AGENCY

Early multimodal evaluation centered on static perception tasks. Seminal works like Visual Question Answering (VQA) (Antol et al., 2015; Hudson & Manning, 2019) and image captioning on datasets like COCO (Chen et al., 2015) were foundational for representation learning. More recent comprehensive platforms, such as MMBench (Liu et al., 2024) and SEED-Bench (Li et al., 2023), aggregated numerous tasks, yet they all share a unifying limitation: their static and non-interactive nature. Models perform single-turn perception on fixed inputs, which fails to evaluate crucial agentic capabilities like sequential decision-making or long-term planning.

### 2.2 INTERACTIVE AGENT BENCHMARKS: AGENCY WITH A MODAL BOTTLENECK

To address the lack of agency, a second wave of benchmarks introduced interactive environments. This evolution began in text-based worlds like Jericho (Hausknecht et al., 2020), expanded to embodied AI in 3D simulators such as AI2-THOR (Kolve et al., 2017) and Habitat (Savva et al., 2019), and extended to grounded language in ALFWorld (Shridhar et al., 2021) and complex digital tasks in WebArena (Zhou et al., 2023) and Mind2Web (Deng et al., 2023). Despite this significant leap towards agency, a prevalent modal bottleneck constrains the majority of these benchmarks, as perception is typically limited to vision and text. Recent game-based works like BALROG (Paglieri et al., 2025) further highlight deep reasoning deficiencies even within these limited modalities. While pioneering platforms like SoundSpaces 2.0 (Chen et al., 2020) incorporated audio for navigation, a comprehensive, diagnostic approach to omni-modality has been missing.

### 2.3 OMNI-MODAL MODELS AND THE EVALUATION CHASM

This long-standing modal bottleneck has recently escalated into a critical evaluation chasm with the arrival of true omni-modal foundation models like Google's Gemini and OpenAI's GPT-4o. These models are natively designed to process a fluid combination of text, image, audio, and video, yet our primary tools for evaluating agency lack the sensory richness to test these new faculties. Current evaluations fail to assess how these powerful models perform in dynamic, multi-sensory scenarios where they must make choices.

OMNIPLAY is designed to bridge this chasm. Unlike recent high-fidelity benchmarks such as SIMA (Team et al., 2024), which aim to simulate realistic open-world interactions to train generalist agents ("flight simulators"), OmniPlay prioritizes *diagnostic control* ("wind tunnels"). By systematically creating tasks requiring synergistic fusion and stress-testing agents with controlled sensory conflicts, OMNIPLAY provides a dedicated diagnostic platform to rigorously evaluate the true interactive and reasoning capabilities of modern omni-modal agents.

## 3 THE OMNIPLAY BENCHMARK

This section details the OmniPlay benchmark. We articulate our design philosophy (Section 3.1), introduce the diagnostic suite (Section 3.2), and validate its design (Section 3.3).

### 3.1 CORE DESIGN PRINCIPLES

Design Philosophy: Wind Tunnel vs. Flight Simulator. Unlike high-fidelity benchmarks focused on open-world realism (Team et al., 2024), OmniPlay prioritizes diagnostic control. Akin to an aerodynamic *wind tunnel* rather than a full *flight simulator*, we simplify physics to isolate specific cognitive mechanisms (e.g., fusion) without confounding variables. To create this controlled environment, we

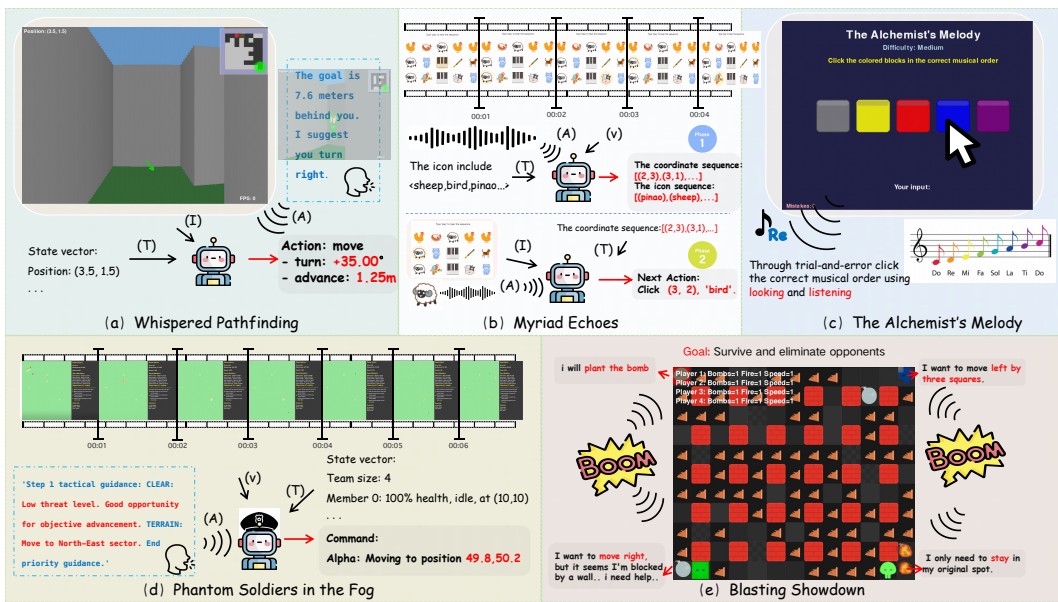

Figure 2: The OmniPlay suite. (a) Visuo-auditory navigation. (b) Sequence replication. (c) Abstract reasoning. (d) Real-time strategy. (e) Multi-agent combat.

adhered to three principles (formalism in Appendix B): (1) Modality Complementarity: Tasks are *unsolvable* without fusing information from disparate channels. (2) Controlled Modality Conflict: We inject conflicting cues to stress-test fusion robustness. (3) Various Modality Complexity: We vary combinations to probe for modality biases.

## 3.2 THE OMNIPLAY DIAGNOSTIC SUITE

OmniPlay is explicitly designed as a diagnostic toolkit where each game acts as a targeted probe for specific mechanisms in a "capability stack." A comprehensive summary table of the games is provided in Appendix C (Table 5).

- **Whispered Pathfinding** *(Fusion Stress Test)*. A 3D maze navigation task requiring the synergy of visual layouts and auditory instructions. **Core Probe:** Its performance under controlled conflict directly measures the robustness of the fusion mechanism itself.

- **Myriad Echoes** *(Perception & Memory Positive Control)*. A high-bandwidth sequence replication task. **Core Probe:** Success here proves perceptual capabilities are intact, allowing us to attribute failures in strategic tasks to higher-order processing rather than basic perception.

- **The Alchemist's Melody** *(Hypothesis Testing & Credit Assignment)*. A rule discovery task. **Core Probe:** Isolates (1) *Hypothesis Testing* (structured exploration) and (2) *Credit Assignment* (linking actions to delayed feedback), distinguishing reasoning from brute-force search.

- **Phantom Soldiers in the Fog** *(Strategic Integration)*. An RTS game requiring planning under uncertainty. **Core Probe:** Measures the ability to *integrate* outputs of lower-level functions (perception, grounding) into long-horizon plans.

- **Blasting Showdown** *(Dynamic Attention Reweighting)*. A multi-agent arena. **Core Probe:** Diagnoses adaptive attentional policies—specifically, whether models can dynamically reweight modality importance (e.g., prioritizing audio cues when vision is insufficient) based on context.

## 3.3 BENCHMARK VALIDATION

To ensure the benchmark's validity and diagnostic effectiveness (addressing Reviewer 1 & 4's concerns), we conducted three quantitative analyses:

**1. Modality Interdependence Validation.** We validated that our tasks require *genuine* cross-modal fusion through systematic ablation. As detailed in Table 1, removing any single modality leads to catastrophic degradation (e.g., $> 2.7\times$ increase in navigation steps), confirming the necessity of multi-sensory integration.

Table 1: Validation of modality interdependence. Performance degrades significantly when a single modality is removed. ($^\star$ denotes degradation $> 2\times$).

| Task (Gemini 2.5 Pro, Hard) | Full | w/o Audio | w/o Image | w/o Text |
|---|---|---|---|---|
| Whispered Pathfinding (Steps↓) | 36.2 | 99.4$^\star$ | 45.9 | 118.9$^\star$ |
| Myriad Echoes (Mean Score↑) | 10.2 | 8.8 | 9.1 | 1.0$^\star$ |

**2. Difficulty Calibration.** We validated our task scaling by analyzing human performance gradients. As shown in Table 2, expert players exhibit expected monotonic degradation patterns across Easy/Medium/Hard levels, confirming our difficulty manipulation is perceptually meaningful.

Table 2: Validation of difficulty calibration via human expert performance gradients. (Lower is better for Steps; higher for Score/SR).

| Task (Human Expert Performance) | Easy | Medium | Hard |
|---|---|---|---|
| Whispered Pathfinding (Steps↓) | 5.2 | 8.3 | 15.6 |
| Myriad Echoes (Score↑)[†] | 3.70 | 2.50 | 2.60 |
| Phantom Soldiers (SR↑) | 1.00 | 0.98 | 0.95 |

[†]The slight rebound on Hard is due to a conservative strategy shift under high memory load.

**3. Conflict Resolution as a Diagnostic Tool.** Crucially, we validated our modality conflict conditions as effective stress tests by comparing AI degradation against human resilience. A pilot study with human experts (N=6) on *Whispered Pathfinding* (Hard) reveals a stark dichotomy (Table 3). Humans demonstrated robust conflict arbitration with only a graceful performance degradation (-6.1%), whereas top AI models exhibited catastrophic failure (-51.6%). This proves the benchmark exposes architectural weaknesses rather than general difficulty.

Table 3: Validation of conflict scenarios as a diagnostic tool (Whispered Pathfinding, Hard). Humans exhibit graceful degradation, while AI shows catastrophic failure.

| Agent | No Conflict | Audio Conflict | % Degradation |
|---|---|---|---|
| Human Expert (N=6 Pilot) | 94.2% | 88.5% | **-6.1%** |
| Gemini 2.5 Pro | 89.4% | 43.3% | **-51.6%** |

AI data from full runs (N=50). Human data from a dedicated pilot study (N=6, 20 trials each).

# 4 EXPERIMENTAL SETUP

This section details the experimental methodology used to evaluate state-of-the-art omni-modal models on the OmniPlay benchmark. We first introduce the models and baselines under evaluation, and then describe our comprehensive evaluation protocol and metrics.

## 4.1 MODELS AND BASELINES

Our evaluation suite comprises six representative omni-modal models, covering both proprietary and open-source ecosystems:

- **Proprietary Models:** Google's **Gemini 2.5 Pro** and **Gemini 2.5 Flash** (Comanici et al., 2025).
- **Open-Source Models: Qwen-2.5-Omni (7B)** (Xu et al., 2025), **MiniCPM-o-2.6 (8B)** (Yao et al., 2024), **Baichuan-Omni-1.5 (7B)** (Li et al., 2025), and **VITA-1.5 (7B)** (Fu et al., 2025).

API restrictions prevented the evaluation of OpenAI's GPT-4o, whose audio-preview version lacks simultaneous multi-modal support.

We contextualize agent performance using two critical baselines. A Random Agent provides a performance floor by uniformly sampling actions. More importantly, we established a robust Human Expert Baseline by recruiting 12 experienced gamers ($>$500 hours), balanced for gender and stratified by age. Participants completed a mandatory familiarization phase (min. 10 warm-up episodes per game) to reach a stable skill plateau. A detailed breakdown of the recruitment protocol and inter-player agreement analysis is provided in Appendix E.

### 4.2 Evaluation Protocol and Metrics

Our primary goal is to diagnose the inherent, foundational capabilities of these models, for which zero-shot evaluation is the established paradigm (Brown et al., 2020). To ensure a fair comparison, every agent is evaluated on a fixed set of evaluation seeds for each task. We employ a hierarchical metric system to balance diagnostic precision with cross-task comparability.

**(1) Task-Specific Raw Metrics.** Each game has a primary raw metric that directly measures performance on its native scale (e.g., navigation steps in *Whispered Pathfinding*). For tasks with multiple objectives, we use a weighted composite score to reflect their relative importance. For example, in *The Alchemist's Melody*, we weight correctness three times higher than exploration efficiency. Full definitions are provided in Appendix D.

**(2) Normalized Performance Score (NPS).** To enable high-level comparison, we normalize raw scores relative to the random (0) and human expert (100) baselines:

$$\text{NPS} = 100 \times \frac{\text{Score}_{\text{model}} - \text{Score}_{\text{random}}}{\text{Score}_{\text{human}} - \text{Score}_{\text{random}}} \tag{1}$$

An NPS of 100 indicates human-level performance, while scores above 100 signify superhuman capabilities.[1]

**Transparency in Reporting.** To ensure full transparency, our main results (Table 4) report *both* NPS and the primary raw score side-by-side. Diagnostic experiments in the appendices primarily use raw metrics to isolate specific phenomena, as unnormalized values are more direct and interpretable for such targeted analyses.

## 5 Results and Analysis

Our experiments reveal critical insights into omni-modal models. We first quantify a performance dichotomy (Section 5.1), then present diagnostic experiments that uncover its mechanistic root cause (Section 5.2).

### 5.1 Quantifying the Dichotomy: Memory as a Control Condition

While the existence of a memory-reasoning performance gap in transformers has been qualitatively observed in unimodal settings, three critical questions have remained unanswered in the omni-modal, interactive regime: (1) What is the quantitative magnitude? (2) Does superhuman unimodal memory transfer to multi-modal streams? (3) What is the mechanistic cause? OmniPlay provides the first systematic answers. Figure 3a quantifies the gap, revealing it is larger than previously suspected. Table 4 presents a focused comparison designed as a controlled experiment.

**Superhuman Memory as a Positive Control.** The memory-intensive task, *Myriad Echoes (Hard)*, is designed as a positive control to validate foundational perceptual and encoding capabilities. Success here is crucial: it demonstrates that models can flawlessly perceive and transcribe high-bandwidth, omni-modal input streams. As expected, top models excel, with **Gemini 2.5 Pro** achieving a 3.9× raw score advantage over the human expert (10.2 vs. 2.6). This confirms the perceptual pipeline is not the bottleneck.

---

[1]As a competitive multi-agent environment (zero-sum PvP), *Blasting Showdown* follows a distinct tournament protocol and is excluded from NPS calculations to maintain statistical rigor.

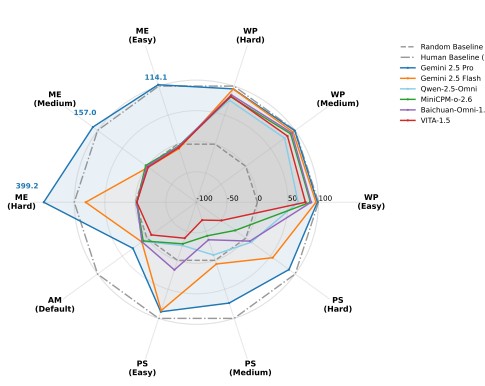

(a) NPS across 10 benchmarked tasks.

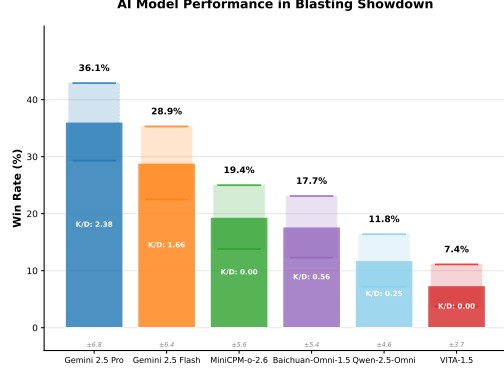

(b) *Blasting Showdown* AI-vs-AI tournament.

Figure 3: Overall performance evaluation. (a) Radar chart across 10 NPS-benchmarked tasks reveals a dichotomy between superhuman memory and sub-par reasoning (error margins: SEM over N=50 episodes). (b) AI-vs-AI tournament in *Blasting Showdown* shows no dominant strategy (error bars: SEM over 50 games).

Table 4: Performance dichotomy on the most challenging memory and strategic reasoning tasks. AI models exhibit superhuman memory but struggle with strategic planning. All metrics: Mean ± SEM over 50 runs.

| Model | Myriad Echoes (Hard) | | Phantom Soldiers (Hard) | |
|---|---|---|---|---|
| | NPS | Raw Score | NPS | Raw Score |
| Gemini 2.5 Pro | $399.2 \pm 3.6$ | 10.2 | $87.5 \pm 3.5$ | 91.6 |
| Gemini 2.5 Flash | $81.4 \pm 4.2$ | 1.9 | $54.5 \pm 4.7$ | 73.5 |
| Qwen-2.5-Omni | $-1.7 \pm 4.9$ | 0.0 | $11.2 \pm 5.8$ | 23.3 |
| MiniCPM-o-2.6 | $-2.5 \pm 5.5$ | 0.0 | $-21.5 \pm 7.0$ | 8.9 |
| **Human Expert** | **100.0** | **2.6** | **100.0** | **98.5** |
| Random Baseline | 0.0 | 0.1 | 0.0 | 17.8 |

Raw scores: Myriad Echoes = mean score; Phantom Soldiers = normalized mission score. Human baseline is mean over N=12 experts. Full details in Appendix D.

**Strategic Failures Attributed to Higher-Order Processing.** With perceptual capabilities validated, we can confidently attribute failures in strategic tasks to higher-order processing. In *Phantom Soldiers (Hard)*, even **Gemini 2.5 Pro** falls short of the human baseline (91.6 vs. 98.5). This scientifically-valid comparison, enabled by our control task, allows us to rule out low-level perception as the cause of failure.

## 5.2 DIAGNOSING THE ROOT CAUSE: BRITTLE MODALITY FUSION

A superficial reading might dismiss our findings as "expected": transformers excel at memory but struggle with reasoning. However, this narrative conflates architectural capacity with empirical behavior in novel regimes. Our diagnostic experiments reveal that the actual mechanisms underlying this gap are neither obvious nor well-understood. We provide converging evidence for a specific, testable hypothesis: catastrophic brittleness in modality fusion is a primary bottleneck that cascades to reasoning failures. Moreover, our experiments uncover several counter-intuitive phenomena that challenge the "expected result" framing.

**Modality Conflict Reveals Foundational Brittleness.** We first stress-tested the models' fusion mechanisms by injecting controlled modality conflicts in *Whispered Pathfinding*. As shown in Figure 4, conflicts induce drastic performance degradation. For **Gemini 2.5 Pro**, audio or text conflicts cause a catastrophic drop in efficiency from 89.4% to 43.3% and 32.2%, respectively. We also observe asymmetrical sensitivity: **Gemini 2.5 Flash** is remarkably resilient to auditory conflicts but highly vulnerable to textual ones, implying a hierarchical reliance on vision and text over audio.

The Counter-Intuitive "Less is More" Paradox. This brittleness helps explain a counter-intuitive "less is more" paradox observed in our modality ablation experiments (Figure 5). While top models like **Gemini 2.5 Pro** require all modalities for synergistic tasks, models with weaker fusion mechanisms paradoxically improve when a sensory channel is removed. For **MiniCPM-o-2.6** in *Whispered Pathfinding*, removing the visual modality boosts its Efficiency Score from 48.8% to 81.4%. This is not an "expected" result; it demonstrates that immature fusion is not merely suboptimal but actively harmful, a critical diagnostic insight.

**Complementary Diagnostic Evidence.** Beyond these core findings, our diagnostic suite reveals several additional weaknesses that converge on the theme of brittle fusion:

*(1) Text-Modality Bias.* Models exhibit a strong preference for textual over non-textual modalities. Substituting auditory alerts with equivalent textual descriptions in *Phantom Soldiers* yielded consistent performance gains across all models (Appendix F.6), suggesting that audio fusion remains fragile.

*(2) Dual-Channel Noise Brittleness.* Visual corruption in *Phantom Soldiers* acts as a *dual-channel attack*—simultaneously degrading spatial information and rendering UI text (e.g., unit status) unreadable. This caused Gemini 2.5 Pro's score to plummet by over 80% (Appendix F.3), suggesting models lack robust intermediate semantic representations and instead learn brittle end-to-end pixel-to-action mappings.

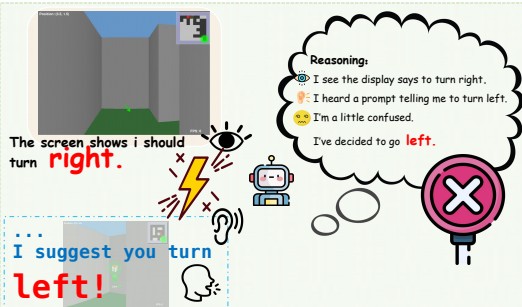

(a) A modality conflict scenario.

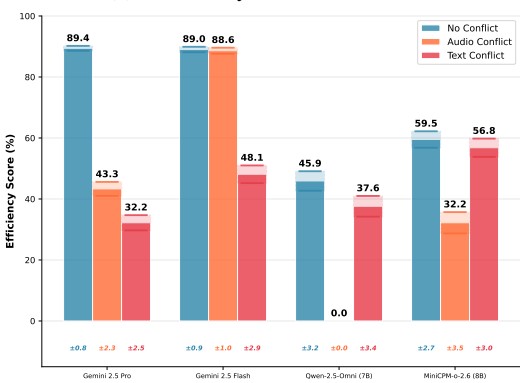

(b) Resulting efficiency scores.

Figure 4: Modality conflict experiments in *Whispered Pathfinding* expose fragile fusion mechanisms.

*(3) Instruction-Following Gap.* This brittleness contrasts sharply with proprietary models' ability to leverage *explicit* textual guidance. In *The Alchemist's Melody*, providing hints boosted Gemini models to 100% completion, while open-source models showed no improvement (Appendix F.4), revealing a significant instruction-following gap.

**Synthesis: From Symptoms to Mechanisms.** Our diagnostic experiments provide converging, multi-modal evidence for a unified mechanistic account: the performance dichotomy stems from catastrophic brittleness in foundational fusion mechanisms, which cascades to higher-order failures. This conclusion is supported by four independent lines of evidence: (1) the 51% efficiency collapse under modality conflict; (2) the paradoxical "less is more" effect; (3) the 80% score drop under dual-channel noise; and (4) the lack of adaptive attention re-weighting. Critically, these findings are not "expected results" but empirical discoveries enabled by OmniPlay's diagnostic design. While architectural priors suggested transformers might struggle, the specific failure mode—brittle fusion—could not have been predicted a priori. Our work transforms vague intuitions ("models can't reason well") into a precise, actionable diagnosis ("fusion brittleness is a primary bottleneck").

## 6 CONCLUSION

We introduced **OmniPlay**, the first interactive benchmark designed to *diagnose*, rather than merely measure, omni-modal agents' fusion and reasoning capabilities. Our evaluation of six leading models revealed a critical dichotomy: *superhuman memory but sub-par strategic reasoning*. We traced this fragility to brittle fusion mechanisms that fail catastrophically under modality conflict and exhibit a counter-intuitive "less is more" paradox.

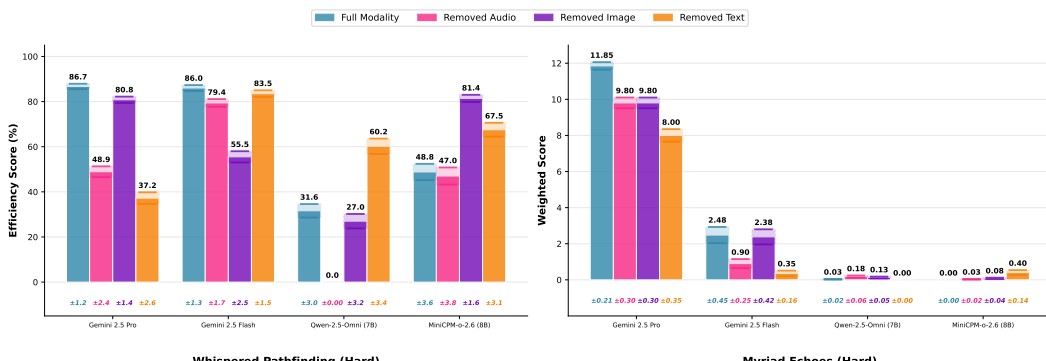

Figure 5: Modality ablation experiments on the 'Hard' difficulty for *Whispered Pathfinding* (left) and *Myriad Echoes* (right). Error margins (SEM) reveal two key phenomena: (1) For top models, removing any modality hurts performance in a synergistic task (necessity). (2) For other models, removing a modality can paradoxically improve performance ("less is more").

Our findings carry a significant implication for the pursuit of AGI: simply scaling models may not be sufficient to bridge the gap to robust, real-world intelligence. The path forward requires a research focus that extends beyond architectural depth to explicitly address the foundational challenges of synergistic fusion, conflict arbitration, and resilient reasoning.

### 6.1 LIMITATIONS AND FUTURE DIRECTIONS

While OmniPlay advances multimodal evaluation, we acknowledge key limitations. (1) Simulated Environments: Our games lack the sensor noise and physics of real-world robotics; validating findings on embodied agents is a critical next step. (2) Zero-Shot Focus: We primarily evaluate zero-shot capabilities. Investigating whether fine-tuning can remedy the observed brittleness—or merely masks deficiencies—remains an open question. (3) Limited Model Coverage: API restrictions excluded GPT-4o; future work should expand coverage. (4) Human Baseline Scale: Our N=12 expert baseline, while reliable, is limited in scale and cultural diversity. (5) Task Diversity: The focus on discrete action spaces leaves continuous control scenarios explored. (6) Language Limitation: The current English-only implementation requires cross-lingual validation to assess generalization.

### 6.2 OPEN-SOURCE COMMITMENT

Despite the limitations, OmniPlay's core diagnostic principles are broadly applicable. To maximize community impact and enable reproducible research, we commit to releasing the complete Omni-Play platform under the **MIT License** upon publication. The release will include:

- All five game environments with full source code.
- Evaluation scripts and metric computation pipelines.
- Pre-computed baseline results and human study protocols.
- Documentation and tutorials for benchmark extension.

We hope this work catalyzes a shift toward more rigorous, mechanistic evaluation of omni-modal intelligence and provides a foundation for the community to extend these diagnostic methods.

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

APPENDIX

## A  USE OF LARGE LANGUAGE MODELS IN MANUSCRIPT PREPARATION

We utilized LLMs as writing assistants during the preparation of this manuscript. Their role was strictly limited to improving grammar, refining phrasing, and enhancing the overall readability and clarity of the text. The conceptualization of the research, the design and execution of the experiments, and the analysis and interpretation of the results are entirely the original work of the authors.

## B  FORMALISM AND DESIGN PRINCIPLES

This appendix provides a detailed description of the generalized Markov Decision Process (MDP) framework used in OmniPlay and illustrates how our core design principles are instantiated within this formalism.

### B.1  GENERALIZED INTERACTION FRAMEWORK

To provide a unified and rigorous description of agent-environment interaction across our diverse suite of games, we model each task within a generalized Markov Decision Process (MDP) framework. This formalism captures the sequential, turn-based nature of our benchmark. The interaction is defined by the primary components $(S, A, T, G, \Omega, O)$:

- $S$: The set of true, underlying world states, which may be fully or partially observable.
- $A$: The agent's action space, which can be discrete, continuous, or hybrid.
- $T$: The state transition function, $T(s'|s, a)$.
- $G$: A set of goal states, $G \subseteq S$.
- $\Omega$: The multi-modal observation space. At each timestep $t$, the agent receives an observation $o_t \in \Omega$ composed of a tuple of available sensory inputs: $o_t = (\mathcal{I}_t, \mathcal{V}_t, \mathcal{A}_t, \mathcal{T}_t)$.
- $O$: The observation function, $O(o_t|s_t)$.

To process the omni-modal observation $o_t$, the agent first employs a set of modality-specific encoders $(E_\mathcal{I}, E_\mathcal{V}, E_\mathcal{A}, E_\mathcal{T})$ to obtain unimodal representations. These representations are then integrated by a fusion module, $\mathcal{F}$, to produce a unified context vector, $c_t$:

$$c_t = \mathcal{F}(E_\mathcal{I}(\mathcal{I}_t), E_\mathcal{V}(\mathcal{V}_t), E_\mathcal{A}(\mathcal{A}_t), E_\mathcal{T}(\mathcal{T}_t)) \tag{2}$$

This context vector $c_t$ is then used to update the agent's internal state or history representation, $h_t$. At each timestep, the agent's policy $\pi(a_t|h_t)$ selects an action $a_t$. The agent's objective is to learn a policy that maximizes the probability of generating a successful trajectory. Let $\tau = (s_0, a_0, s_1, a_1, \dots)$ denote a trajectory. The probability of observing $\tau$ given a policy $\pi$ is:

$$P(\tau|\pi) = p(s_0) \prod_{t=0}^{|\tau|-1} \pi(a_t|h_t)T(s_{t+1}|s_t, a_t) \tag{3}$$

Let $\mathcal{T}_G$ be the set of all trajectories that terminate in a goal state $s_g \in G$. The optimal policy $\pi^*$ is the one that solves:

$$\pi^* = \arg\max_\pi \sum_{\tau \in \mathcal{T}_G} P(\tau|\pi) \tag{4}$$

### B.2  FORMALIZING THE CORE DESIGN PRINCIPLES

Our three core design principles—Modality Complementarity, Controlled Modality Conflict, and Various Modality Complexity—are not merely abstract concepts but are formally embedded within the MDP structure described in Appendix B.1.

**Modality Complementarity (formerly Interdependence).** This principle is primarily realized through the design of the state transition function $T(s'|s, a)$ and the goal states $G$. A task embodies complementarity if, for many states $s$, there is no single modality in the observation $o_t$ that provides sufficient information for the policy $\pi$ to choose an action $a_t$ that maintains a high probability of reaching $G$. Formally, let $\pi_m$ be a policy that only conditions on a single modality $m \in \{\mathcal{I}, \mathcal{V}, \mathcal{A}, \mathcal{T}\}$. A task with strong complementarity ensures that:

$$\max_{\pi} \sum_{\tau \in \mathcal{T}_G} P(\tau|\pi) > \max_m \left( \max_{\pi_m} \sum_{\tau \in \mathcal{T}_G} P(\tau|\pi_m) \right) \tag{5}$$

This inequality formally states that the performance of a full omni-modal policy is significantly greater than the best possible uni-modal policy.

**Controlled Modality Conflict.** We introduce conflict by manipulating the observation function $O(o_t|s_t)$. In a conflict scenario, the observation $o_t = (\ldots, m_i, \ldots, m_j, \ldots)$ contains information from two or more modalities, $m_i$ and $m_j$, that suggest contradictory optimal actions. For instance, modality $m_i$ suggests an action $a_i$ that maximizes the value function $V^\pi(s)$, while modality $m_j$ suggests an action $a_j$ that leads to a much lower value. This forces the agent's fusion module $\mathcal{F}$ and policy $\pi$ to resolve the ambiguity.

**Various Modality Complexity.** This principle is reflected in the diversity of the observation spaces $\Omega$ and action spaces $A$ across our suite of five games. For example, the $\Omega$ for *Whispered Pathfinding* contains continuous spatialized audio, while the $\Omega$ for *The Alchemist's Melody* involves discrete auditory tones. Similarly, the action space $A$ ranges from continuous navigation controls to discrete clicking actions. This variation across the set of MDPs $\{\text{MDP}_1, \ldots, \text{MDP}_5\}$ ensures that we are not testing a model's specialization to a single type of environment but its general omni-modal capability.

## C   GAME ENVIRONMENT DETAILS

This appendix provides detailed descriptions for each of the five game environments in the OmniPlay suite. For each game, we outline its core objective, the modalities and user interface (UI) presented to the agent, its core gameplay mechanics, and the prompting structure. Screenshots of each game's UI and prompts are included for visual reference.

Table 5 provides a comprehensive summary of the five game environments, detailing their core objectives, modality interplay, and the specific capabilities they are designed to probe.

### C.1   WHISPERED PATHFINDING

**Core Objective.** The agent's goal is to navigate a procedurally generated 3D maze to find a hidden, stationary target location.

**Modalities and UI.** The agent perceives the environment through three primary modalities. An example of the UI is shown in Figure 6.

- **Image (I):** A first-person visual feed showing the maze walls and corridors.

- **Audio (A):** Synthesized verbal guidance delivered as Text-to-Speech audio. An example of the transcribed audio content is shown in Figure 7.

- **Text (T):** The complete turn-based prompt, which provides a structured dump of the agent's current state and tasks the agent with generating the next action.

Table 5: Overview of the OmniPlay Diagnostic Game Suite. (Moved from Main Text). Modalities: I=Image, V=Video, A=Audio, T=Text. Difficulties: E=Easy, M=Medium, H=Hard.

| Game | Core Objective & Modality Interplay | Modalities | Difficulties | Key Diagnostic Probe |
|---|---|---|---|---|
| **Whispered Pathfinding** | Navigate a 3D maze by **synergizing** visual pathways with auditory instructions. **Conflict** is introduced when audio commands contradict visual cues. | I, A, T | E, M, H | **Synergistic Fusion** and **Conflict Resolution**. |
| **Myriad Echoes** | **Transcribe** a complex audio-visual sequence, then **execute** it on a static grid. | P1: V, A, T P2: I, A, T | E, M, H | **Cross-Modal Grounding** and **Working Memory**. |
| **The Alchemist's Melody** | Discover a **latent mapping** between colors and musical notes through trial-and-error. | I, A, T | Medium | **Abstract Reasoning** and emergent rule learning. |
| **Phantom Soldiers in the Fog** | Command a squad under a "fog of war" by **integrating** asynchronous, multi-source information. | V, A, T | E, M, H | **Strategic Planning under Uncertainty**. |
| **Blasting Showdown** | Survive in a competitive arena by **reacting** to crucial off-screen auditory cues. | I, A, T | N/A | **Reactive Strategy** and reliance on non-dominant modalities. |

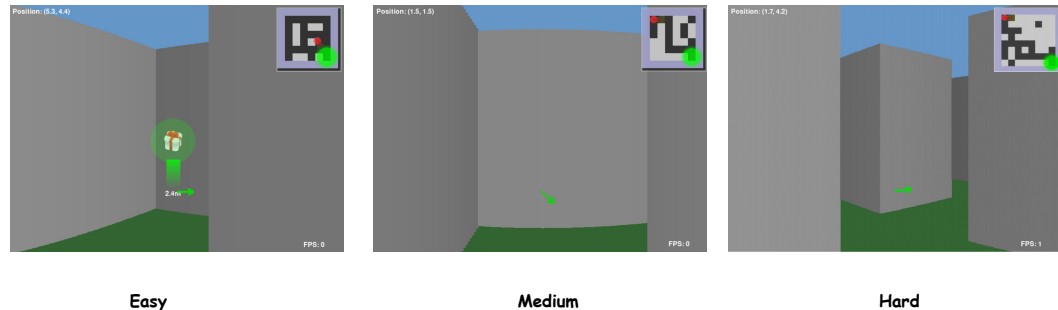

Easy                          Medium                          Hard

Figure 6: User interface for the *Whispered Pathfinding* environment across difficulties.

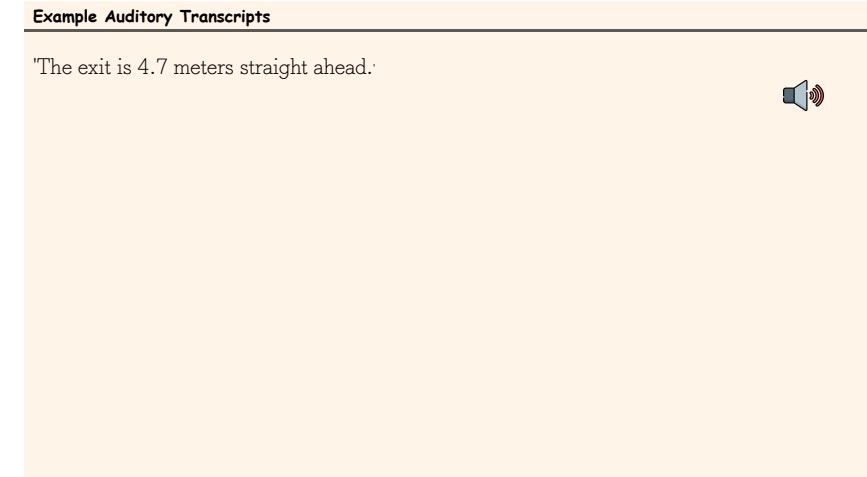

Figure 7: Example Auditory Transcript from *Whispered Pathfinding*. This text content is converted to speech for the agent.

**Gameplay Mechanics.** The agent's action space is continuous, consisting of rotation and forward movement. Success requires synergizing the visual information with the auditory guidance.

### C.1.1 PROMPTING STRUCTURE

Interaction with the agent is structured via a system prompt that defines its role and a turn prompt that provides state information for each action.

**System Prompt.** The system prompt, shown in Figure 8, is used to initialize the agent's behavior, defining its persona, capabilities, and required output format.

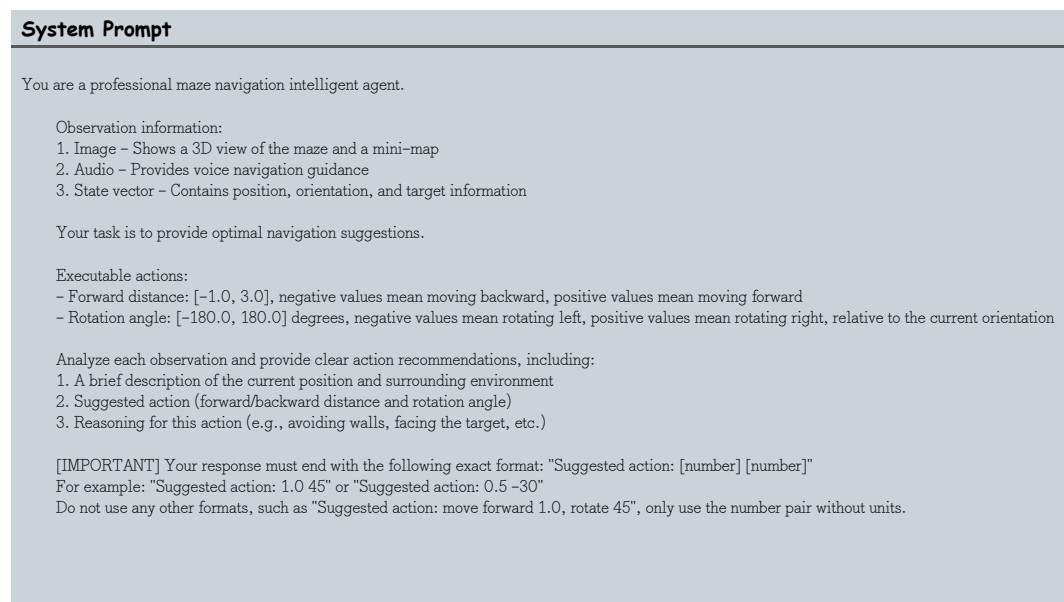

Figure 8: System Prompt for *Whispered Pathfinding*.

**Turn Prompt.** At each decision step, the text modality consists of the prompt shown in Figure 9, where {state_description} is populated with real-time data.

**Turn Prompt**

Please analyze the current maze environment and provide navigation suggestions.

Environment state information:
{state_description}

Please provide the following:
1. Environment analysis: Describe the current position, orientation, and relationship to the target position
2. Suggested action: Provide specific forward distance and rotation angle
3. Navigation rationale: Explain why you chose this action

Remember to end your response with the format "Suggested action: [forward distance] [rotation angle]".
For example: "Suggested action: 1.0 –45"

```
state_description = (
    f"Current position: x={}, y={}"
    f"Current orientation: {}°"
    f"Distance to target: {}m"
    f"Direction to target: {}°"
    "Distance to walls: "",
)
```

Figure 9: Turn Prompt for *Whispered Pathfinding*.

## C.2 MYRIAD ECHOES

**Core Objective.** This task diagnoses the full perception-to-symbol-to-action pipeline across two distinct phases.

**Modalities and UI.** The UI for both phases is shown in Figure 10.

- **Phase 1 (Transcription):** The agent is presented with a dynamic sequence of highlighted icons (**Video**) and corresponding unique sounds (**Audio**).

- **Phase 2 (Execution):** The agent is presented with a static grid of icons (**Image**) and receives auditory feedback (**Audio**) on clicks. The ground-truth sequence is provided via a textual prompt (**Text**).

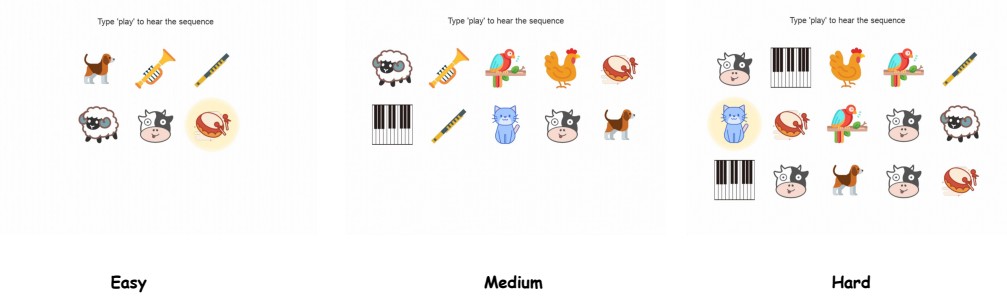

Figure 10: User interface for the *Myriad Echoes* environment across difficulties.

**Gameplay Mechanics.** In Phase 1, the agent must parse the multi-modal stream. In Phase 2, it must execute the parsed sequence by clicking the icons in the correct order.

### C.2.1 PROMPTING STRUCTURE

**System Prompt.** The agent is initialized with the system prompt shown in Figure 11.

**System Prompt**

You are a professional AI assistant for a sound–based memory game.

Game Rules:
1. The game first plays an audiovisual sequence where each icon lights up and plays a corresponding sound.
2. Your task is to remember the order of the sequence.
3. Then, repeat the sequence by clicking the icons in the same order.
4. Icons include animals (dog, cat, bird, cow, sheep, chicken) and musical instruments (piano, trumpet, drum, flute).

Input Information:
1. Video − shows the sequence being played, with icons lighting up in order.
2. Audio − plays the sound associated with each icon in the sequence.
3. Screenshot − shows the current layout of the icons on the game interface.

Your Task:
1. Watch the video and listen to the audio to memorize the order and position of each icon in the sequence.
2. Analyze the game interface screenshot to identify the position of each icon.
3. Based on your memory of the sequence, provide the coordinates for which icon should be clicked next.

Coordinate System:
– Icons are arranged in a grid, starting from the top–left corner.
– Rows and columns are both 1–indexed.
– For example: the icon in the first row and first column has the coordinate (1, 1).
– The icon in the second row and third column has the coordinate (2, 3).

Figure 11: System Prompt for *Myriad Echoes*.

**Turn Prompt.** The prompt for Phase 2 is the ground-truth sequence, visualized in Figure 12 and 12.

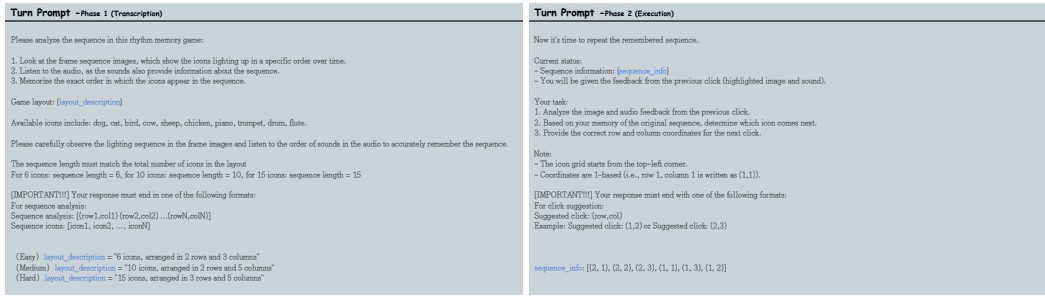

Figure 12: Turn Prompt and UI for Phase 1 (left) and Phase 2 (right) of *Myriad Echoes*.

## C.3 THE ALCHEMIST'S MELODY

**Core Objective.** The agent must discover a latent mapping between colors and musical notes to reproduce a specified musical scale.

**Modalities and UI.** The UI is shown in Figure 13.

- **Image (I):** A set of clickable colored blocks.

- **Audio (A):** Clicking a block plays a musical note.

- **Text (T):** A highly structured, real-time state dump containing feedback, sequence status, and strategic hints.

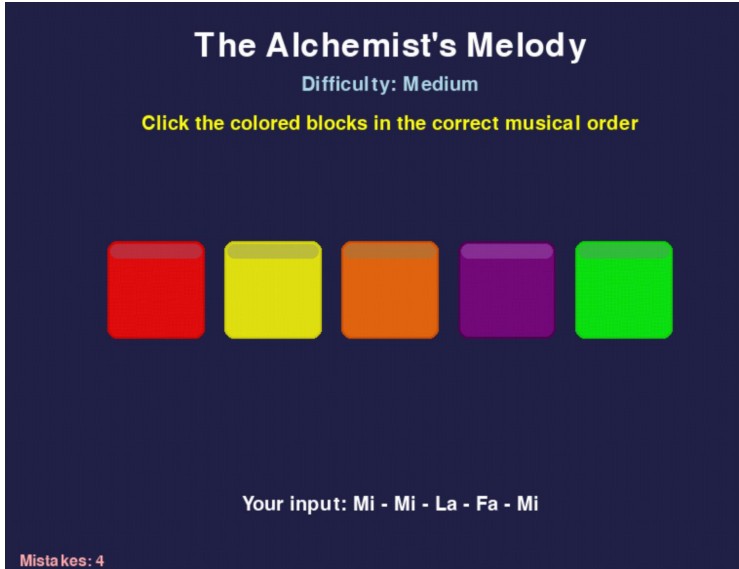

The Alchemist's Melody

Figure 13: User interface for *The Alchemist's Melody*.

**Gameplay Mechanics.** The color-note mapping is randomized per episode. The agent must deduce it via trial-and-error, guided by the rich textual feedback.

### C.3.1 PROMPTING STRUCTURE

**System Prompt.** The agent's role is defined by the system prompt shown in Figure 14.

---

**System Prompt**

"You are a MULTIMODAL AI agent playing a musical color-matching game.\n"
"## ROLE\n"
"Click exactly ONE coloured block per turn to reproduce the target melody.\n"
"\n"
"## GAME RULES\n"
"1. Musical order (ascending): do → re → mi → fa → sol → la → si.\n"
"2. At the start of each round, the FIRST note is chosen at random; it may be any notes.\n"
"3. After the first note, you must continue in the same ascending order **without skipping any note** until the melody is complete (wrap around if needed).\n"
"4. After any wrong click, the sequence resets to this round's first note.\n"
"5. Colour-to-note mapping is RANDOMIZED **each round**; learn it anew from feedback.\n"
"\n"
"## OBSERVATION FIELDS\n"
"• `image` − current board frame (colours & highlights).\n"
"• `audio` − sound from **your previous click**.\n"
"• `currently_in_correct_sequence` (bool)\n"
"• `needs_restart_from_beginning` (bool)\n"
"• `current_correct_sequence` (list of colours already correct)\n"
"• `input_length` (int)\n"
f"{available_colors_info}\n"
"*Clicking any other colour is invalid.*\n"
"*The order of these colors has no significance; it's completely random.*\n"
"\n"
"## DECISION CHECKLIST\n"
"1. If `needs_restart_from_beginning` is true → restart with this round's first note.\n"
"2. Otherwise pick the next consecutive note based on `current_correct_sequence`—do **not** skip any note.\n"
"3. Identify the NOTE you just heard by pairing your last action with the `audio` feedback.\n"
"4. Choose the colour that plays the required next note.\n"
"\n"
"## OUTPUT FORMAT\n"
"Reply with **ONLY** two uppercase tokens separated by a comma and a space:\n"
"<COLOUR>, <NOTE>\n"
"• <COLOUR> e.g. `BLUE`.\n"
"• <NOTE> ∈ {DO, RE, MI, FA, SOL, LA, SI}.\n"
"No other text, punctuation, or line breaks."

available_colors_info =
'Available Color Blocks in this round:\n− BLUE\n− GREEN\n−
RED\n− GREY\n− YELLOW'

---

Figure 14: System Prompt for *The Alchemist's Melody*.

**Turn Prompt.** The agent receives a composite prompt including the task instruction and the detailed game state, as shown in Figure 15.

**Turn Prompt**

Current game observation and detailed state from environment:

{detailed_game_state}
Based on the detailed game state above, what color block should I click next?
– If currently_in_correct_sequence is True: Continue the musical sequence
– If needs_restart_from_beginning is True: Start from the beginning note
– If currently_in_correct_sequence is False: Choose a different color than the last clicked one
{conversation_context}

Remember to follow the ascending musical order without skipping notes.

detailed_game_state =
"DETAILED GAME STATE (from environment):\n\nLAST ACTION INFO:\n– Last Clicked Action ID: 0\n– Last Clicked Block Color: \n– Previous Block Color (last_last): \n\nSEQUENCE STATUS (CRITICAL FOR DECISION):\n– Currently in Correct Sequence: False\n– Needs Restart from Beginning: True\n– Current Correct Sequence: []\n– Previous Clicks History: []\n– Sequence Length: 5\n– Current Input Length: 0\n\nGAME STATUS:\n– Game Over: False\n– Current Tick: 0.0\n– Attempts: 0\nSTRATEGY HINTS:\n– If 'current_correct_sequence' has items: These are the correct colors so far\n– If 'previous_clicks' shows history: Learn from past click patterns\n– Use sequence position to determine next required musical note"

conversation_context = 'RECENT HISTORY:\nRound 1:\n  Game State: Currently in correct sequence=False\n  Action: RED\n', \nRound 2:\n.....

Figure 15: Turn Prompt for *The Alchemist's Melody*, showing the structured state representation.

### C.4 PHANTOM SOLDIERS IN THE FOG

**Core Objective.**   The agent acts as an RTS commander for a squad, aiming to achieve strategic objectives under a "fog of war."

**Modalities and UI.**   The UI is shown in Figure 16.

- **Video (V):** A top-down tactical map.

- **Text (T):** Mission objectives and unit status reports.

- **Audio (A):** Structured tactical guidance delivered as Text-to-Speech audio.  An example transcript is shown in Figure 17.

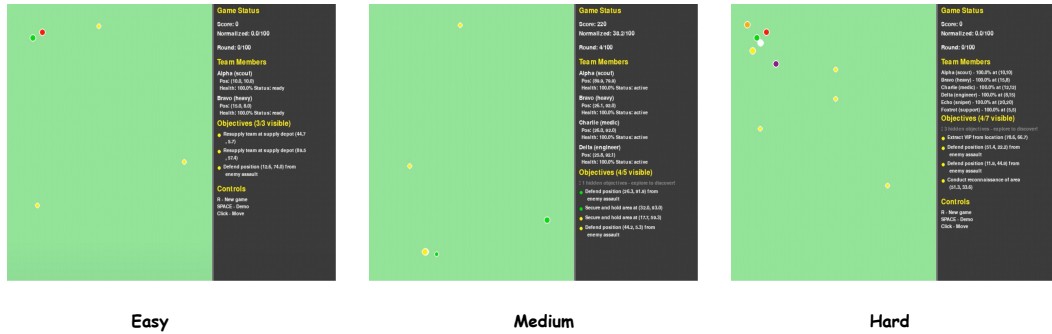

Easy                                    Medium                                    Hard

Figure 16: User interface for *Phantom Soldiers in the Fog* across difficulties.

Figure 17: Example Auditory Transcript from *Phantom Soldiers in the Fog*.

**Gameplay Mechanics.** The agent issues high-level commands. Success hinges on integrating visual, textual, and structured audio-channel guidance.

C.4.1 PROMPTING STRUCTURE

**System Prompt.** The extensive system prompt, shown across three parts in Figure 18, defines the complex role of the agent.

**System Prompt**

You are commanding a military team in a cooperative mission. You MUST provide EXACTLY ONE command per turn.

FORBIDDEN: Multiple commands like "COMMAND: 0 move 20 30" AND "COMMAND: 1 recon 40 50"
CORRECT: Only one command like "COMMAND: 0 move 20 30"

If you provide more than one command, the system will ERROR and use a default command instead.

KEY GAME MECHANICS:

{command_reliability_note}

HIDDEN OBJECTIVES:
– Some objectives are HIDDEN and not visible initially
– You must EXPLORE different areas to discover hidden objectives
– Scout team members have higher discovery probability (80% vs 40%)
– Send scouts to unexplored areas to find new objectives
– Discovery hints may indicate "unusual activity" in areas with hidden objectives

MOVEMENT UNCERTAINTY:
– Team members DO NOT move to exact coordinates you specify
– Movement has ERROR based on:
  * Role precision (Scout: low error, Heavy: high error)
  * Health status (injured = more error)
  * Movement distance (longer moves = more error)
– Expect actual positions to deviate from your targets
– Plan for imprecise movement in your strategy

{info_sources}

**System Prompt**

STRATEGIC CONSIDERATIONS:
– Balance exploration (finding hidden objectives) vs completion (finishing known objectives)
– Use scouts for exploration and discovery
– Account for movement errors in positioning
– Monitor team health and status for optimal assignment
– Hidden objectives may have high score values – worth discovering!

COMMAND FORMAT – PROVIDE EXACTLY ONE OF THESE:

**Individual Command (one member):**
COMMAND: [member_id] [action] [x] [y]

**Team Command (all members together):**
COMMAND: all [action] [x] [y]

**Multi–member Command (specific members together):**
COMMAND: 0,1,2 [action] [x] [y]

**Available Actions:** move, attack, defend, recon, status
**Coordinates:** x, y: 0–100 (actual position will vary due to movement error)

EXAMPLES OF CORRECT RESPONSES:
"Based on the current situation, I'll send the scout to explore. COMMAND: 0 recon 25 30"
"The team should move together to the objective. COMMAND: all move 45 20"
"Two scouts should explore this area. COMMAND: 0,1 recon 70 80"

EXAMPLES OF INCORRECT RESPONSES (WILL CAUSE ERRORS):
"COMMAND: 0 move 25 30" followed by "COMMAND: 1 recon 45 20"
Multiple command lines in any form
Suggesting multiple commands for "efficient coordination"

**System Prompt**

FINAL REMINDER: ONE COMMAND ONLY!
– Analyze the situation thoroughly
– Choose the SINGLE most important action
– Provide exactly ONE command
– Plan step–by–step across multiple turns, not all at once

Provide your strategic analysis, then end with exactly ONE command.

command_reliability_note:
'\n    COMMAND EXECUTION:\n
– Commands execute deterministically – all valid commands will succeed\n
– Focus on strategic positioning and optimal task assignment\n
– No need to account for random command failures\n\n'

info_sources :
'    INFORMATION PROVIDED:\n–
Vector: Team member states (health, status, position) + global info (rounds remaining, normalized score)\n
– Video: Visual sequence showing game state progression and team member movements over time\n
– Discovery hints: Clues about nearby hidden objectives'

Figure 18: System Prompt for *Phantom Soldiers in the Fog* (Parts 1, 2, and 3).

**Turn Prompt.** At each step, the agent receives the turn prompt shown in Figure 19.

---

**Turn Prompt**

Current game state:
{state_desc}

     CRITICAL REMINDER: EXACTLY ONE COMMAND ONLY!

You MUST provide exactly ONE command in your response. Multiple commands will cause SYSTEM ERRORS!

    DO NOT DO THIS: Provide multiple "COMMAND:" lines
    DO THIS: Provide exactly one "COMMAND:" line

Choose the SINGLE most important action for this turn. You can plan additional moves for future turns.

Available inputs:
– Vector: Team member states (health, status, position) + global info (rounds remaining, normalized score)
– Video: Visual sequence showing game state progression
– Audio: Tactical guidance and team communications
– Discovery hints: Clues about nearby hidden objectives

Analyze the situation and provide your ONE command.

state_desc:
'Team size: 2\nMember 0: 100% health, idle, at (10,10)\nMember 1: 100% health, idle, at (15,8)\nRounds remaining: 100\nScore: 0.0/100\nVideo: Game state video sequence available'

---

Figure 19: Turn Prompt for *Phantom Soldiers in the Fog*.

C.5    BLASTING SHOWDOWN

**Core Objective.** Four agents compete in a destructible arena to be the last one standing.

**Modalities and UI.** The UI is shown in Figure 20.

- **Image (I):** A top-down view of the arena.

- **Text (T):** Status updates on all players.

- **Audio (A):** Crucial sound cues (e.g., bomb placements) essential for survival.

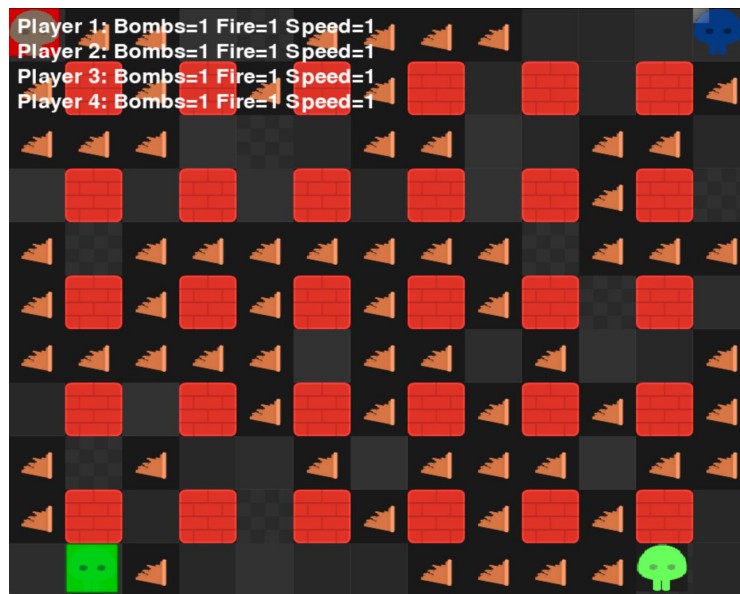

**Blasting Showdown**

Figure 20: User interface for the *Blasting Showdown* environment.

**Gameplay Mechanics.** Inspired by Bomberman, agents move and place bombs. Auditory cues are designed to be critical for reacting to off-screen threats.

C.5.1 PROMPTING STRUCTURE

**System Prompt.** The agent's competitive persona is set by the system prompt in Figure 21.

---

**System Prompt**

You are an AI player playing Bomberman game, playing as the {color_name} character (Player {player_id+1}).
You need to make intelligent decisions based on current game state information, game screen images, and sound events.

Game Rules:
1. You can move on the map or place bombs
2. Bombs create cross-shaped explosions that can destroy soft walls and hit players
3. Soft walls (brown blocks) have a chance to drop power-ups when destroyed: increase fire power (explosion range), increase bomb count, or improve movement speed
4. Players hit by flames will be eliminated, your goal is to defeat other players and survive as much as possible
5. The last surviving player wins

Map Elements:
– Empty space: Can move freely
– Soft walls (brown): Can be destroyed by bombs
– Hard walls (gray): Cannot be destroyed or passed through
– Bombs: Will explode after being placed, creating cross-shaped flames
– Flames: Will damage players and destroy soft walls
– Power-ups: Enhance player abilities

After analyzing images, sounds, and game state information, make the best decision:
1. Move to safe positions, avoid being hit by bombs
2. Strategically place bombs to destroy soft walls or defeat opponents
3. Collect valuable power-ups to enhance abilities
4. Predict opponent actions and react accordingly

Please return your decision in JSON format:
For movement: {{"action_type": 0, "target_x": <target x coordinate>, "target_y": <target y coordinate>}}, for example: {{"action_type": 0, "target_x": 1, "target_y": 2}} but please ensure the target coordinates are within map boundaries and do not exceed maximum movement distance.
For placing bomb: {{"action_type": 1, "target_x": 0, "target_y": 0}}

Ensure the return format strictly follows requirements, only return one valid JSON object, do not add other explanatory text.

color_name = ["Red", "Blue", "Green", "Yellow"]
player_id = ["0", "1", "2", "3"]

---

Figure 21: System Prompt for *Blasting Showdown*.

**Turn Prompt.** The turn prompt, shown in Figure 22, provides comprehensive state information and varies depending on whether the agent is the active player.

**Turn Prompt**

Current game state analysis – Round {obs['step']}:

You are Player{player_id+1}, {"Alive" if player_info['alive'] == 1 else "Dead"}
Your position: ({position_x}, {position_y})
Your attributes:
– Fire power: {player_info['fire_power']} (bomb explosion range)
– Bomb count: {player_info['bomb_count']} (maximum simultaneous bombs)
– Currently placed bombs: {player_info['active_bombs']}
– Movement speed: {player_info['speed']}
– Trapped status: {"Yes" if player_info['trapped'] == 1 else "No"}

{"    WARNING: You are currently in bomb explosion range! Evacuate immediately!" if player_in_danger else ""}

Movement limitations:
– Your maximum movement distance is {max_move_distance} tiles (Manhattan distance)
– This is base distance (5 tiles) plus speed attribute bonus (speed value – 1)
– You cannot pass through walls or bombs – bombs become obstacles after placement
– If target position exceeds movement range, you will move to the farthest reachable point

Other players' positions:
{chr(10).join(other_players) if other_players else "No other surviving players"}

Danger zone warnings:
{chr(10).join([f"    Bomb at position({x},{y}), {timer} steps until explosion, fire range {fire} tiles, will affect horizontal area from ({x–fire},{y}) to ({x+fire},{y}) and vertical area from
({x},{y–fire}) to ({x},{y+fire})!"
    for i, (x, y, timer, owner, fire) in enumerate([(obs['state']['bombs']['positions_x'][i],
            obs['state']['bombs']['positions_y'][i],
            obs['state']['bombs']['countdown'][i],
            obs['state']['bombs']['owner'][i],
            obs['state']['bombs']['fire_power'][i])
             for i in range(obs['state']['bombs']['count'])])]) if bombs else "Currently no bomb threats on the field"}

**Turn Prompt**

Important reminder: Bombs become obstacles after placement and cannot be passed through! Consider this when planning routes.

{state_changes}

{game_events_description}

{history_summary}

Please analyze the attached game screen image and sound events, assess the current situation, and decide whether to move to a safe position, place a bomb, or collect power-ups.
Prioritize safety! Stay away from bomb explosion zones, especially those with short countdowns.
Return a JSON action in the correct format, for example {{"action_type": 0, "target_x": 5, "target_y": 3}} to move to position (5,3).

e.g.
State changes:\n– Bomb count on field increased: 0 → 1'
game_events_description = 'No special game events this turn.
Recent action history:\n– Round 0: move to (12,14)

Figure 22: Turn Prompt for *Blasting Showdown*, for an active player (left) and an observing player (right).

# D    EXPERIMENTAL PARAMETERS AND METRICS

This appendix provides a comprehensive breakdown of the experimental parameters, evaluation protocols, and metrics used in our study to ensure full reproducibility.

## D.1    MODEL AND API PARAMETERS

For all proprietary models (**Gemini 2.5 Pro** and **Gemini 2.5 Flash**), we utilized the official, latest stable API versions available at the time of evaluation. For all models, both proprietary and open-source, we used their default decoding parameters (e.g., for temperature, top-p, and top-k) as provided by their respective APIs or standard inference scripts, without any model-specific tuning.

## D.2 EVALUATION EPISODES AND SEEDS

Our evaluation protocol is built upon a fixed set of evaluation seeds for each task, ensuring that every agent is evaluated on the exact same sequence of game scenarios. The number of episodes and seeds used for each game is detailed in Table 6. For the NPS-benchmarked tasks, each human expert played 10 episodes for each difficulty level.

Table 6: Number of evaluation episodes per task for AI and random agents.

| Game | Difficulty | Seeds / Episodes |
|---|---|---|
| *Whispered Pathfinding* | Easy, Medium, Hard | 50 |
| *Myriad Echoes* | Easy, Medium, Hard | 50 |
| *The Alchemist's Melody* | Medium | 50 |
| *Phantom Soldiers in the Fog* | Easy, Medium, Hard | 30 |
| *Blasting Showdown* | N/A (AI vs. AI) | 50 games |

## D.3 PRIMARY EVALUATION METRICS

Our primary metric for cross-task comparison is the **Normalized Performance Score (NPS)**, as defined in the main text. The raw 'Score' used in the NPS calculation is derived from a custom, task-specific scoring function for each game. The following section details these scoring functions and other diagnostic metrics.

## D.4 TASK-SPECIFIC SCORING AND DIAGNOSTIC METRICS

We designed a unique set of metrics for each game to capture nuances of agent performance. Table 7 provides a high-level overview of the metrics collected for each game. The subsequent paragraphs detail the specific scoring functions used for NPS calculation.

**Whispered Pathfinding.** For this navigation task, the **final score for NPS is based on the inverse of 'Mean Steps (Trimmed)'**, as fewer steps indicate higher performance. This trimmed mean is calculated after removing the highest and lowest step counts to reduce outlier impact.

**Myriad Echoes.** The **final score for NPS is a weighted sum**: 50% from 'Mean Score' (execution phase), 25% from 'Mean Coordinate Accuracy' (parsing phase), and 25% from 'Mean Icon Accuracy' (parsing phase).

**The Alchemist's Melody.** This task evaluates abstract reasoning. The **final score for NPS is the 'Score' metric**, a composite calculated from multiple performance facets as detailed in Table 8.

**Phantom Soldiers in the Fog.** This RTS task uses two final metrics. The **final score for NPS is a weighted sum: 50% from 'Success Rate' and 50% from 'Normalized Score'**.

- **Success Rate:** The ratio of completed objectives to total objectives, representing mission completion.
- **Normalized Score:** A score from 0-100 reflecting tactical and strategic efficiency, detailed below.

### D.4.1 DETAILED CALCULATION OF THE NORMALIZED SCORE

The Normalized Score is derived from several components:

1. **Main Score** ($S_{main}$): Sum of points from all completed objectives.
2. **Auxiliary Score** ($S_{aux}$): A bonus for efficient command execution.
3. **Max Possible Score** ($S_{max}$): This theoretical ceiling is calculated to normalize performance. It includes a base score ($S_{base}$), plus bonuses for efficiency ($B_{efficiency}$) and

Table 7: Overview of diagnostic metrics collected for each game in the OmniPlay suite.

| Game Environment | Collected Metrics |
|---|---|
| *Whispered Pathfinding* | Mean/Min/Max Steps, Mean Invalid Actions, Mean Steps (Trimmed), **NPS Score (Inverse of 'Trimmed Mean Steps')** |
| *Myriad Echoes* | Success Rate, Mean Score (Execution), Mean Coordinate Accuracy (Parsing), Mean Icon Accuracy (Parsing), Parsing Failure Rate, **NPS Score (Weighted Sum)** |
| *The Alchemist's Melody* | Completion Rate, Composite Score, **NPS Score ('Composite Score')** |
| *Phantom Soldiers in the Fog* | Success Rate, Normalized Score, **NPS Score (Weighted Sum of 'Success Rate' and 'Normalized Score')** |
| *Blasting Showdown* | Win Rate, Total Kills, Total Deaths, K/D Ratio |

Table 8: Composite score calculation for *The Alchemist's Melody*.

| Component | Formula |
|---|---|
| A: Hit Rate | Hit Rate $\times$ 30 |
| B: Step Efficiency | 30 if steps $\leq$ required, else penalized |
| C: Correct Streak | (Total Correct Streak Length / Total Steps) $\times$ 10 |
| D: Error Penalty | 10 - (Total Error Streak Length / Total Steps) $\times$ 10 |
| E: Color Error Penalty | 15 - (Same-Color Error Length / Total Steps) $\times$ 15 |
| F: Exploration | (Color Changes / (Steps - 1)) $\times$ 5 |
| **Total Score** | **Sum of A + B + C + D + E + F** |

completing the mission within an optimal number of rounds ($R_{opt}$) relative to the maximum allowed rounds ($R_{max}$). For the Hard difficulty, a dynamic bonus ($B_{dynamic}$) is also added.

$$S_{base} = \sum S_{obj} \text{ for all objectives}$$
$$B_{rounds} = S_{base} \times (1 - R_{opt}/R_{max}) \times 0.5$$
$$B_{efficiency} = S_{base} \times 0.3$$
$$S_{max} = S_{base} + B_{rounds} + B_{efficiency} + B_{dynamic}$$

4. **Optimal Rounds** ($R_{opt}$): This is a complex heuristic estimating the minimum rounds required. It accounts for visible objectives, rounds needed to discover hidden objectives (factoring in scout units), and an overhead for map exploration.

5. **Final Normalized Score** ($S_{norm}$): The final score is a dynamic normalization of the main and auxiliary scores, considering efficiency relative to optimal rounds. The base formula is $S_{norm} = \min(100, \max(0, (S_{main}/S_{max}) \times 100))$.

**Blasting Showdown.** As a competitive multi-agent game, this task does not use NPS. Performance is measured with metrics from a 50-game tournament.

# E HUMAN BASELINE VALIDATION PROTOCOL

This appendix details the protocol used to establish a reliable, representative, and diverse Human Expert baseline, which is critical for the calculation of the Normalized Performance Score (NPS). Our methodology was designed to ensure fairness, stability, and high inter-player agreement while incorporating greater demographic diversity.

**Participant Recruitment and Demographics.** We recruited a total of **12 human participants**, balanced for gender and stratified by age. A key criterion for selection remained extensive gaming experience, with all participants reporting over 500 hours of gameplay across various genres relevant to the tasks in OmniPlay. Participants were compensated for their time.

The cohort was structured as follows:

- **Young Adult Group (n=8):** Ages 20-35, consisting of 4 male and 4 female participants (mean age 25.5).
- **Middle-Aged Adult Group (n=4):** Ages 35-50, consisting of 2 male and 2 female participants (mean age 41.0).

This stratified sampling aimed to provide a more robust baseline by capturing potential variations in cognitive strategies and reaction times across different age groups and genders.

**Familiarization and Training Protocol.** To ensure that the collected data represented expert-level, stable performance rather than a learning phase, each participant underwent a mandatory warm-up and training protocol for every game environment. Before any data was recorded for a specific game (including its different difficulty levels), each participant was required to play a minimum of 10 non-recorded 'warm-up' episodes. The purpose of this phase was twofold: first, to allow participants to fully familiarize themselves with the game's unique user interface, controls, and objectives; second, to allow their performance and strategies to stabilize and reach a performance plateau. Our experimenters verbally confirmed with each participant that they felt confident in their understanding of the task before proceeding to data collection.

**Data Collection and Analysis.** Following the warm-up phase, each participant played a set number of recorded episodes for each task, as detailed in Table 9. The final 'Human Expert' score reported in the main text for each task is the mean score calculated across all episodes from **all 12 participants**. This larger and more diverse sample provides a statistically more stable estimation of the human performance baseline.

Table 9: Number of recorded evaluation episodes per human participant.

| Game Environment | Episodes per Participant |
|---|---|
| *Whispered Pathfinding* | 10 |
| *Myriad Echoes* | 10 |
| *The Alchemist's Melody* | 10 |
| *Phantom Soldiers in the Fog* | 10 |

**Inter-Player Reliability.** A critical aspect of validating our human baseline is ensuring high agreement among the expert players. The detailed statistics of our human expert performance, including mean raw scores and inter-player standard deviation (SD), are presented in **Table 22**. As shown, the overall SD remained consistently low relative to the mean score across most tasks, confirming a high degree of agreement on optimal strategies.

Further analysis of the inter-player variance revealed logical patterns tied to our diverse participant pool. On tasks emphasizing cognitive skills with clear optimal solutions, such as *Myriad Echoes* and *The Alchemist's Melody*, the performance difference between age groups and genders was minimal, resulting in a low SD (typically 5-10% of the mean score). For tasks requiring strategy and efficient navigation like *Phantom Soldiers* and *Whispered Pathfinding*, we observed a moderate increase in variance, with the younger participant group (20-35 years) generally achieving slightly higher efficiency scores. The highest variance was observed in *Blasting Showdown* (raw data not in NPS tables), a task heavily reliant on reaction speed and mechanical skill. In this game, younger male participants tended to achieve the highest performance, aligning with common patterns in competitive gaming. This controlled and interpretable variance, even with our demographically diverse sample, confirms that our collected baseline is stable and representative of expert human performance.

**Limitations of the Human Baseline.** We acknowledge that while our human baseline of 12 participants (6 male, 6 female; mean age 30.7, SD 7.6)[2] provides balance in age and gender, it still

---

[2]The reported mean age and standard deviation are calculated from the average age of each group (25.5 for the young adult group and 41.0 for the middle-aged adult group).

Table 10: Overview of all diagnostic experiments conducted in this study.

| Diagnostic Test Type | Target Game(s) | Methodology / Intervention | Capability Probed |
|---|---|---|---|
| **Modality Conflict** | *Whispered Pathfinding* | Introduce semantic contradictions between modalities (e.g., audio command vs. visual cue; textual state vs. other cues). | Fusion Robustness, Conflict Resolution, Modality Bias |
| **Modality Ablation** | *Whispered Pathfinding*, *Myriad Echoes* | Systematically remove one modality (audio, image, or text) at a time and evaluate performance on the remaining subset. | Modality Interdependence, Synergistic Fusion, "Less is More" Paradox |
| **Robustness to Noise** | *Phantom Soldiers in the Fog* | Inject noise into sensory inputs: corrupting audio-channel text with random words/letters; applying visual noise (Gaussian, salt-pepper) to the map. | Perceptual Robustness, Generalization beyond clean data |
| **Aided Reasoning via Prompting** | *Myriad Echoes*, *The Alchemist's Melody* | Augment the turn prompt with explicit, helpful information (e.g., current sequence step, learned color-note mapping). | Advanced Instruction Following, Knowledge Application |
| **Task Simplification** | *Myriad Echoes* | Remove the second (execution) phase of the task, converting it into a single-phase perception-to-symbol transcription task. | Benchmark Complexity Validation, Disentangling Perception vs. Action |
| **Modality Substitution** | *Phantom Soldiers in the Fog* | Replace information from one modality with its semantic equivalent in another (e.g., audio alerts replaced with textual alerts). | Modality-Agnostic Representation, Modality Preference/Bias |

has limitations regarding cultural background and broader cognitive diversity. Future work could explore larger-scale and more varied cross-cultural evaluations to further generalize the human performance benchmark.

## F    DETAILED DIAGNOSTIC EXPERIMENTS

This appendix provides the detailed methodologies and full results for the diagnostic experiments summarized in the main text. These experiments were designed to probe specific capabilities and failure modes of the evaluated omni-modal agents. Table 10 provides a high-level overview of all diagnostic tests conducted.

Table 11: Full results for Modality Conflict experiments on *Whispered Pathfinding*. Performance is measured by Mean Steps (lower is better).

| Model | Easy Difficulty | | | | | Medium Difficulty | | | | | Hard Difficulty | | | | |
|---|---|---|---|---|---|---|---|---|---|---|---|---|---|---|---|
| | Mean | Min | Max | Invalid | Trimmed | Mean | Min | Max | Invalid | Trimmed | Mean | Min | Max | Invalid | Trimmed |
| *Baseline (No Conflict)* | | | | | | | | | | | | | | | |
| gemini-2.5-pro | 7.6 | 5 | 10 | 0.0 | 7.6 | 10.2 | 7 | 14 | 0.0 | 10.1 | 42.6 | 13 | 152 | 0.0 | 36.2 |
| gemini-2.5-flash | 16.1 | 4 | 70 | 0.8 | 10.9 | 23.2 | 6 | 87 | 1.2 | 19.0 | 43.5 | 18 | 112 | 2.7 | 37.15 |
| qwen-2.5-omni | 70.3 | 10 | 273 | 29.5 | 52.5 | 64.2 | 11 | 132 | 28.1 | 62.4 | 130.1 | 32 | 253 | 52.6 | 128.2 |
| MiniCPM-o-2.6 | 27.0 | 7 | 73 | 7.5 | 23.8 | 34.4 | 6 | 86 | 10.7 | 31.5 | 110.8 | 34 | 255 | 35.8 | 99.5 |
| *Audio Conflict* | | | | | | | | | | | | | | | |
| gemini-2.5-pro | 22.4 | 8 | 71 | 0.1 | 18.1 | 27.1 | 10 | 108 | 0.0 | 21.8 | 140.6 | 58 | 244 | 0.2 | 133.7 |
| gemini-2.5-flash | 20.7 | 4 | 122 | 1.7 | 10.1 | 35.1 | 8 | 95 | 2.1 | 28.6 | 57.8 | 28 | 127 | 0.8 | 38.0 |
| qwen-2.5-omni | 93.4 | 27 | 154 | 37.9 | 94.1 | 142.6 | 18 | 500 | 54.3 | 113.5 | 240.4 | 103 | 500 | 91.2 | 225.1 |
| MiniCPM-o-2.6 | 48.0 | 11 | 126 | 15.5 | 42.9 | 76.4 | 27 | 143 | 23.8 | 70.7 | 172.4 | 41 | 427 | 57.1 | 157.0 |
| *Text Conflict* | | | | | | | | | | | | | | | |
| gemini-2.5-pro | 39.9 | 17 | 99 | 0.1 | 32.6 | 49.2 | 18 | 149 | 0.0 | 43.0 | 160.1 | 40 | 295 | 0.0 | 157.2 |
| gemini-2.5-flash | 11.0 | 5 | 21 | 1.0 | 10.0 | 17.5 | 14 | 20 | 0.0 | 18.0 | 126.8 | 9 | 267 | 6.1 | 123.6 |
| qwen-2.5-omni | 52.9 | 24 | 115 | 21.8 | 48.8 | 69.5 | 21 | 187 | 30.8 | 60.9 | 150.8 | 38 | 261 | 50.6 | 145.6 |
| MiniCPM-o-2.6 | 27.0 | 7 | 47 | 7.7 | 27.0 | 86.7 | 27 | 230 | 26.2 | 76.2 | 114.8 | 29 | 137 | 26.1 | 105.2 |

## F.1 MODALITY CONFLICT

To stress-test the robustness of agent's fusion mechanisms, we conducted modality conflict experiments in the *Whispered Pathfinding* environment. We systematically created scenarios where information from different modalities was semantically contradictory, forcing the agent to resolve ambiguity. The full results, compared against the no-conflict baseline, are consolidated in Table 11.

**Audio-Visual Conflict.** In this condition, the visual cues (e.g., on-screen arrow) and textual state information were correct, but the synthesized verbal command was manipulated to suggest a contradictory action (e.g., the visual arrow points right, while the audio says "turn left"). The results in Table 11 show a universal degradation in performance. All models took significantly more steps to solve the maze compared to the baseline, exposing the fragility of their fusion mechanisms. For instance, on the Hard difficulty, Gemini 2.5 Pro's trimmed mean steps increased from 36.2 to 133.7, a nearly fourfold increase in inefficiency, demonstrating that even top-tier models struggle to resolve such conflicts and often follow the misleading audio cue.

**Text-Visual/Audio Conflict.** In this condition, the visual and auditory cues remained correct, but the structured text in the Turn Prompt was manipulated to be misleading by inverting the agent's orientation and the direction to the target. The data reveals a fascinating asymmetrical sensitivity, strongly supporting our main-text conclusion. Gemini 2.5 Pro is severely impacted by this conflict, with its mean steps increasing dramatically across all difficulties. Conversely, Gemini 2.5 Flash appears to almost entirely ignore the misleading text, showing performance that is much closer to the baseline and, on Easy/Medium difficulties, even better than its performance under audio conflict. This strongly suggests an internal modality hierarchy where Gemini 2.5 Flash prioritizes visual and auditory cues, while Gemini 2.5 Pro may have a stronger bias toward structured textual data, making it more vulnerable to this specific type of conflict.

## F.2 MODALITY ABLATION

To investigate the necessity of each modality and uncover potential "less is more" phenomena, we conducted modality ablation studies on *Whispered Pathfinding* and *Myriad Echoes*. In these experiments, we evaluated agent performance under four conditions: the baseline with all modalities ('Full Modality'), and three ablation conditions where either the audio, image, or text modality was removed.

### F.2.1 WHISPERED PATHFINDING

The full results for modality ablation on *Whispered Pathfinding* are presented in Table 12.

Table 12: Full results for modality ablation on *Whispered Pathfinding*. Performance is measured by Mean Steps (lower is better).

| Model | Easy Difficulty | | | | | Medium Difficulty | | | | | Hard Difficulty | | | | |
|---|---|---|---|---|---|---|---|---|---|---|---|---|---|---|---|
| | Mean | Min | Max | Invalid | Trimmed | Mean | Min | Max | Invalid | Trimmed | Mean | Min | Max | Invalid | Trimmed |
| *Full Modality (Baseline)* | | | | | | | | | | | | | | | |
| gemini-2.5-pro | 7.6 | 5 | 10 | 0.0 | 7.6 | 10.2 | 7 | 14 | 0.0 | 10.1 | 42.6 | 13 | 152 | 0.0 | 36.2 |
| gemini-2.5-flash | 16.1 | 4 | 70 | 0.8 | 10.9 | 23.2 | 6 | 87 | 1.2 | 19.0 | 43.5 | 18 | 112 | 2.7 | 37.15 |
| qwen-2.5-omni | 70.3 | 10 | 273 | 29.5 | 52.5 | 64.2 | 11 | 132 | 28.1 | 62.4 | 130.1 | 32 | 253 | 52.6 | 128.2 |
| MiniCPM-o-2.6 | 27.0 | 7 | 73 | 7.5 | 23.8 | 34.4 | 6 | 86 | 10.7 | 31.5 | 110.8 | 34 | 255 | 35.8 | 99.5 |
| *Removed Audio* | | | | | | | | | | | | | | | |
| gemini-2.5-pro | 13.7 | 4 | 80 | 0.1 | 10.5 | 22.4 | 5 | 92 | 0.1 | 19.6 | 123.7 | 14 | 476 | 0.6 | 99.4 |
| gemini-2.5-flash | 9.3 | 4 | 26 | 0.8 | 8.7 | 18.5 | 5 | 51 | 1.4 | 17.4 | 49.9 | 13 | 115 | 2.9 | 48.3 |
| qwen-2.5-omni | 91.5 | 12 | 329 | 35.5 | 82.8 | 108.8 | 13 | 326 | 43.1 | 102.0 | 189.9 | 79 | 424 | 73.6 | 181.1 |
| MiniCPM-o-2.6 | 40.0 | 9 | 177 | 6.8 | 34.1 | 139.0 | 11 | 376 | 29.5 | 125.4 | 108.6 | 31 | 235 | 25.9 | 102.5 |
| *Removed Image* | | | | | | | | | | | | | | | |
| gemini-2.5-pro | 8.5 | 4 | 42 | 0.0 | 7.7 | 22.0 | 6 | 164 | 0.0 | 14.1 | 47.1 | 9 | 130 | 0.0 | 45.9 |
| gemini-2.5-flash | 20.6 | 4 | 269 | 1.1 | 14.4 | 38.1 | 9 | 198 | 1.9 | 34.7 | 95.5 | 21 | 292 | 5.3 | 88.3 |
| qwen-2.5-omni | 42.2 | 11 | 84 | 14.9 | 41.1 | 85.9 | 19 | 219 | 33.7 | 82.4 | 156.1 | 45 | 429 | 63.6 | 135.9 |
| MiniCPM-o-2.6 | 24.6 | 8 | 60 | 9.0 | 22.2 | 49.7 | 17 | 107 | 21.5 | 46.6 | 55.0 | 23 | 97 | 12.0 | 45.0 |
| *Removed Text* | | | | | | | | | | | | | | | |
| gemini-2.5-pro | 34.5 | 11 | 115 | 0.0 | 31.4 | 147.3 | 13 | 461 | 0.0 | 111.4 | 124.2 | 45 | 241 | 0.0 | 118.9 |
| gemini-2.5-flash | 9.2 | 4 | 22 | 1.8 | 9.0 | 34.1 | 5 | 371 | 7.6 | 23.1 | 42.9 | 6 | 120 | 6.7 | 41.5 |
| qwen-2.5-omni | 29.1 | 10 | 49 | 8.2 | 29.0 | 53.8 | 11 | 133 | 23.7 | 49.2 | 89.3 | 22 | 228 | 30.2 | 80.4 |
| MiniCPM-o-2.6 | 19.0 | 6 | 53 | 5.3 | 18.2 | 29.8 | 11 | 58 | 8.7 | 28.6 | 70.6 | 34 | 114 | 13.8 | 68.3 |

**Analysis of 'Removed Audio'.** Removing the audio cues had a universally negative impact on performance, dramatically increasing the number of steps required for all models across all difficulties. This is particularly evident on the Hard difficulty, where, for instance, Gemini 2.5 Pro's trimmed mean steps skyrocketed from 36.2 to 99.4. This result strongly validates the principle of *Modality Interdependence* for this task, as it confirms that the auditory channel provides critical, non-redundant information for efficient navigation that cannot be compensated for by vision and text alone.

**Analysis of 'Removed Image'.** The results from removing the visual modality are particularly revealing. For top-performing models like Gemini 2.5 Pro, performance degrades, though less severely than when audio is removed. This suggests that while vision is helpful, the audio and text cues can still guide the agent effectively. However, for models with weaker fusion mechanisms, we observe a striking "less is more" paradox. On the Hard difficulty, MiniCPM-o-2_6's performance dramatically *improves* when the visual modality is removed, with its mean steps dropping from 110.8 to 55.0. This suggests that for this model, the visual input acts as a 'distractor', and removing it simplifies the decision-making process, leading to a better outcome.

**Analysis of 'Removed Text'.** Removing the textual state information also led to performance degradation, especially for Gemini 2.5 Pro on Medium and Hard difficulties. This indicates that top-tier models effectively ground the coordinate information to their visual perception to plan more efficient routes. Interestingly, for Gemini 2.5 Flash, removing text has a less severe impact and in some cases (Easy/Medium) even slightly improves performance compared to the baseline, suggesting it relies less on explicit coordinate data.

### F.2.2 WHISPERED PATHFINDING

The full results for modality ablation on *Whispered Pathfinding* are presented in Table 12.

### F.2.3 MYRIAD ECHOES

The full results for modality ablation on *Myriad Echoes* are presented in Table 13.

**Analysis of Ablation Conditions.** For this memory- and parsing-intensive task, the results show a more complex pattern. For the top-performing Gemini 2.5 Pro, removing any single modality leads to a severe drop in performance across all metrics (Success Rate, Mean Score, etc.), especially on

Table 13: Full results for modality ablation on *Myriad Echoes* across all difficulties.

| Model | Easy Difficulty | | | | | Medium Difficulty | | | | | Hard Difficulty | | | | |
|---|---|---|---|---|---|---|---|---|---|---|---|---|---|---|---|
| | Succ.(%) | M.Score | Coord. | Icon | ParseF(%) | Succ.(%) | M.Score | Coord. | Icon | ParseF(%) | Succ.(%) | M.Score | Coord. | Icon | ParseF(%) |
| *Full Modality (Baseline)* | | | | | | | | | | | | | | | |
| MiniCPM-o-2.6 | 0 | 0.1 | 0.2 | 0.2 | 50 | 0 | 0.1 | 0.1 | 0.3 | 20 | 0 | 0 | 0 | 0 | 40 |
| gemini-2.5-flash | 0 | 0 | 0 | 0 | 0 | 0 | 0 | 0 | 0 | 0 | 0 | 1.9 | 4.5 | 1.6 | 0 |
| gemini-2.5-pro | 70 | 4.7 | 4.8 | 4.8 | 0 | 10 | 4 | 5.7 | 5.6 | 0 | 60 | 10.2 | 13.5 | 13.5 | 0 |
| qwen-2.5-omni | 0 | 0.1 | 0.1 | 0.3 | 60 | 0 | 0 | 0 | 0 | 50 | 0 | 0 | 0 | 0.1 | 60 |
| *Removed Audio* | | | | | | | | | | | | | | | |
| MiniCPM-o-2.6 | 0 | 0.3 | 0.4 | 0.2 | 50 | 0 | 0.1 | 0.1 | 0 | 20 | 0 | 0 | 0 | 0.1 | 40 |
| gemini-2.5-flash | 0 | 0 | 0 | 0 | 0 | 0 | 0 | 0 | 0 | 0 | 0 | 0.3 | 1.5 | 1.5 | 0 |
| gemini-2.5-pro | 70 | 4.7 | 5.8 | 5.8 | 0 | 0 | 1.9 | 3.4 | 3.4 | 0 | 40 | 8.8 | 10.8 | 10.8 | 0 |
| qwen-2.5-omni | 0 | 0.3 | 0.4 | 0.4 | 20 | 0 | 0 | 0 | 0.1 | 40 | 0 | 0.2 | 0.2 | 0.1 | 10 |
| *Removed Image* | | | | | | | | | | | | | | | |
| MiniCPM-o-2.6 | 0 | 0.3 | 0.3 | 0.1 | 60 | 0 | 0.2 | 0.2 | 0.1 | 30 | 0 | 0.1 | 0.1 | 0 | 20 |
| gemini-2.5-flash | 0 | 0.2 | 0.2 | 0 | 0 | 0 | 0.3 | 1 | 1 | 0 | 0 | 1.7 | 3.1 | 3.1 | 0 |
| gemini-2.5-pro | 30 | 3.2 | 3.7 | 3.7 | 0 | 30 | 5.1 | 7.7 | 7.7 | 0 | 50 | 9.1 | 10.5 | 10.5 | 0 |
| qwen-2.5-omni | 0 | 0.3 | 0.5 | 0.6 | 10 | 0 | 0 | 0 | 0.3 | 30 | 0 | 0.1 | 0.1 | 0.2 | 40 |
| *Removed Text* | | | | | | | | | | | | | | | |
| MiniCPM-o-2.6 | 0 | 0 | 0 | 0 | 60 | 0 | 0.1 | 0.1 | 0 | 40 | 0 | 0.1 | 0.4 | 0.4 | 0 |
| gemini-2.5-flash | 0 | 0 | 0 | 0 | 0 | 0 | 0.2 | 2 | 0 | 0 | 0 | 0.1 | 1.2 | 0 | 0 |
| gemini-2.5-pro | 0 | 0.6 | 3.6 | 3.6 | 0 | 0 | 1.1 | 9 | 9 | 0 | 0 | 1 | 15 | 15 | 0 |
| qwen-2.5-omni | 0 | 0 | 0 | 0 | 40 | 0 | 0.3 | 0.4 | 0.2 | 60 | 0 | 0 | 0 | 0 | 50 |

Table 14: Performance on *Phantom Soldiers in the Fog* (Medium) under different noise conditions.

| Model | Baseline (No Noise) | | Audio Noise | | Image Noise | |
|---|---|---|---|---|---|---|
| | Score | Succ. Rate | Score | Succ. Rate | Score | Succ. Rate |
| gemini-2.5-pro | 78.81 | 0.880 | 46.6 | 0.850 | 14.2 | 0.275 |
| gemini-2.5-flash | 31.20 | 0.570 | 20.5 | 0.483 | 2.9 | 0.050 |
| qwen-2.5-omni | 23.74 | 0.465 | 21.7 | 0.533 | 0 | 0 |
| MiniCPM-o-2.6 | 11.60 | 0.200 | 14.2 | 0.290 | 9.6 | 0.185 |

higher difficulties. This underscores that its superhuman ability is contingent on successfully fusing information from the entire multi-modal stream. For other models, the performance is already very low in the baseline condition, making the impact of ablation less pronounced. However, we can observe that for weaker models, removing the audio or image modality can sometimes slightly reduce the 'Parse Failure Rate', suggesting that a simpler set of inputs, even if incomplete, is less likely to confuse their parsing mechanisms.

## F.3 ROBUSTNESS TO SENSORY NOISE

We investigated the models' resilience to non-ideal sensory inputs by introducing noise into the visual and auditory modalities in the *Phantom Soldiers in the Fog* environment (Medium difficulty). The full results are presented in Table 14.

**Audio Noise Injection and Analysis.** For the audio modality, our goal was to simulate a noisy communication channel by corrupting the transcribed tactical guidance text before speech synthesis. We randomly inserted meaningless 'noise words' and 'noise letters' into the original guidance sentences (Figure 23). The results show that while audio noise degrades performance, most models exhibit a degree of resilience. The impact of this semantic-level audio noise was less severe than the visual noise.

**Image Noise Injection and Analysis.** For the visual modality, we simulated sensor degradation by applying a combination of Gaussian noise, salt-and-pepper noise, and a slight blurring filter to the video feed (Figure 24). The impact was universally catastrophic, with all models experiencing a severe drop in performance.

**Detailed Analysis: The Dual-Channel Nature of Visual Noise.** Our visual corruption in *Phantom Soldiers* constitutes a *compound perturbation*, affecting multiple information channels simultaneously:

**Example of Noisy Audio**

Original:
"Step 5 tactical guidance: WARNING High threat environment detected. FORMATION Spread out to reduce risk. "

Add noise to the audio:
"Step 5 [ding] tactical guidance: WARNING [zap] High threat [chirp] environment detected. [snap] FORMATION Spread [w] out to [b] reduce risk. "

noise_words: ["xyz", "qwe", "abc", "def", "ghi", "jkl", "mno", "pqr", "stu", "vwx",
        "beep", "buzz", "hiss", "static", "crackle", "pop", "zap", "whir",
        "blip", "chirp", "ding", "ping", "click", "snap", "thud", "bang",
        "noise", "audio", "signal", "freq", "wave", "echo", "reverb", "gain" ]

noise_letters: ["x", "z", "q", "j", "k", "v", "w", "y", "p", "f", "g", "h", "b", "n", "m"]

Figure 23: Illustration of audio noise injection via corruption of the source text in *Phantom Soldiers in the Fog*.

**Example of Noisy Image**

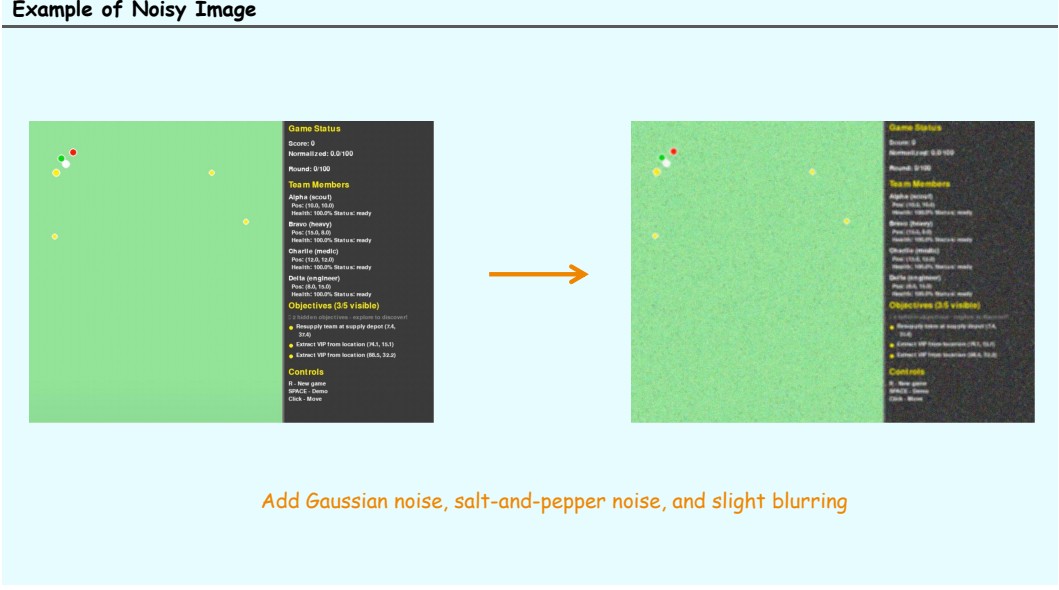

Add Gaussian noise, salt-and-pepper noise, and slight blurring

Figure 24: Illustration of visual noise injection in *Phantom Soldiers in the Fog*.

**(1) Spatial Information Degradation.** The noise corrupts the tactical map's spatial structure, obscuring unit positions and terrain features.

**(2) Textual Information Corruption.** More critically, the same noise renders UI text elements (e.g., unit status, mission objectives) largely unreadable, effectively severing a key information channel.

**Mechanistic Hypothesis.** The catastrophic ¿80% performance drop for Gemini 2.5 Pro suggests that models learn end-to-end pixel-to-action mappings without constructing robust intermediate semantic representations. This is supported by two key observations: (a) **Text-as-Image Processing:** The inability to tolerate noisy text suggests models process UI text via brittle, OCR-like pattern matching, rather than through a robust semantic pathway. (b) **Lack of Cross-Modal Compensa-**

**tion:** Audio cues remained intact under visual noise, yet models failed to compensate by increasing reliance on this channel, indicating rigid, non-adaptive fusion mechanisms.

**Comparison to Human Resilience.** Informal pilot trials (N=3) suggest humans maintain 60-70% of their baseline performance under similar noise conditions. They achieve this by actively shifting attention to the intact audio channel and using prior game knowledge to infer occluded information. This stark contrast underscores the brittleness of current AI fusion.

**Implications for Future Work.** These findings suggest that improving robustness requires architectural innovations beyond scaling, such as explicit fusion modules with learned attention re-weighting under corruption, diverse noise augmentation during training, and modular architectures with separate, robust semantic extraction stages.

### F.4 AIDED REASONING VIA PROMPTING

In contrast to penalizing models with noisy or conflicting data, these experiments investigate their ability to leverage helpful, explicit guidance provided in textual prompts.

**Myriad Echoes.** In the standard version of this task, the agent must internally track its progress through the execution sequence. In the 'Aided Reasoning' version, we augmented the Turn Prompt for Phase 2 with an explicit status update, telling the agent which step of the sequence it was currently on, as shown in Figure 25. The goal was to test if this explicit state information could reduce errors in long-sequence execution. The results, presented in Table 15, show that this hint significantly benefited top-performing models, while weaker models were unable to effectively utilize this information.

---

**Turn Prompt**

```
# ...existing text..

    Current status:
    – This is step {current_step + 1} in the sequence.

# ...existing text..

current_step = n ∈ {0, 1, …, num_icons − 1}
```

Figure 25: Augmented Turn Prompt for *Myriad Echoes*, providing the agent with its current step in the sequence.

**The Alchemist's Melody.** This game requires the agent to deduce a color-note mapping. In the 'Aided Reasoning' condition, we made this task significantly easier by directly providing the agent with its currently learned 'color-note mapping' within the Turn Prompt, as shown in Figure 26. The results in Table 16 are striking. Proprietary models demonstrated a remarkable ability to utilize this hint, jumping to 100% completion. In stark contrast, all tested open-source models failed to leverage this explicit information, highlighting a significant gap in advanced instruction-following and rule-application capabilities.

Table 15: Performance comparison on *Myriad Echoes* with and without aided reasoning prompts.

| Model | Easy Difficulty | | | | | Medium Difficulty | | | | | Hard Difficulty | | | | |
|---|---|---|---|---|---|---|---|---|---|---|---|---|---|---|---|
| | Succ.(%) | M.Score | Coord. | Icon | ParseF(%) | Succ.(%) | M.Score | Coord. | Icon | ParseF(%) | Succ.(%) | M.Score | Coord. | Icon | ParseF(%) |
| *Baseline* | | | | | | | | | | | | | | | |
| gemini-2.5-pro | 70 | 4.7 | 4.8 | 4.8 | 0 | 10 | 4.0 | 5.7 | 5.6 | 0 | 60 | 10.2 | 13.5 | 13.5 | 0 |
| gemini-2.5-flash | 0 | 0 | 0 | 0 | 0 | 0 | 0 | 0 | 0 | 0 | 0 | 1.9 | 4.5 | 1.6 | 0 |
| qwen-2.5-omni | 0 | 0.1 | 0.1 | 0.3 | 60 | 0 | 0 | 0 | 0 | 50 | 0 | 0 | 0 | 0.1 | 60 |
| MiniCPM-o-2.6 | 0 | 0.1 | 0.2 | 0.2 | 50 | 0 | 0.1 | 0.1 | 0.3 | 20 | 0 | 0 | 0 | 0 | 40 |
| *With Aided Prompt* | | | | | | | | | | | | | | | |
| gemini-2.5-pro | 90 | 5.5 | 5.5 | 5.5 | 0 | 70 | 7.0 | 7.0 | 7.0 | 0 | 80 | 12.0 | 12.0 | 10.7 | 0 |
| gemini-2.5-flash | 0 | 0 | 0 | 0 | 0 | 10 | 1.0 | 1.0 | 1.0 | 0 | 0 | 1.0 | 1.0 | 1.5 | 0 |
| qwen-2.5-omni | 0 | 0.5 | 0.3 | 0 | 60 | 0 | 0.4 | 0.3 | 0.2 | 40 | 0 | 0 | 0 | 0 | 60 |
| MiniCPM-o-2.6 | 0 | 0.2 | 0.4 | 0.3 | 20 | 0 | 0.2 | 0.2 | 0.2 | 0 | 0 | 0 | 0 | 0 | 30 |

---

**Turn Prompt**

```
# ...existing text..

    Learned Color–Note Mapping (use this to make informed decisions):
    {learned_color_note_mapping}
    The order of these colors has no significance; it's completely random.

# ...existing text..

learned_color_note_mapping:
'Grey' ='Unknown'
'Blue' ='do'
'Orange' ='Unknown'
'Green' ='mi'
'Yellow' ='fa'
```

Figure 26: Augmented Turn Prompt for *The Alchemist's Melody*, providing the agent with its learned color-note mapping.

Table 16: Performance comparison on *The Alchemist's Melody* with and without aided reasoning prompts.

| Model | Baseline | | With Aided Prompt | |
|---|---|---|---|---|
| | Score | Comp. Rate | Score | Comp. Rate |
| gemini-2.5-pro | 43.154 | 20% | 73.104 | 100% |
| gemini-2.5-flash | 32.048 | 0% | 62.096 | 100% |
| MiniCPM-o-2.6 | 30.294 | 0% | 32.798 | 0% |
| VITA-1.5 | 20.010 | 0% | 18.896 | 0% |
| Baichuan-Omni-1.5 | 31.823 | 0% | 33.548 | 0% |
| qwen-2.5-omni | 31.234 | 0% | 32.722 | 0% |

**Analysis of the Instruction-Following Gap.** The failure of open-source models to utilize explicit hints in *The Alchemist's Melody* likely stems from several interrelated factors:

- **Model Scale and Cognitive Capacity:** Smaller models (7-8B) may lack the capacity to simultaneously process multi-modal perceptions, internalize abstract textual rules from the prompt, and apply these rules to ongoing decision-making. Proprietary models, with substantially larger parameter counts, have more capacity for this complex multi-tasking.

Table 17: Performance comparison on *Myriad Echoes* between the original task and the simplified (perception-only) task.

| Model | Easy Difficulty | | | | | Medium Difficulty | | | | | Hard Difficulty | | | | |
|---|---|---|---|---|---|---|---|---|---|---|---|---|---|---|---|
| | Succ.(%) | M.Score | Coord. | Icon | ParseF(%) | Succ.(%) | M.Score | Coord. | Icon | ParseF(%) | Succ.(%) | M.Score | Coord. | Icon | ParseF(%) |
| *Original Task (Baseline)* | | | | | | | | | | | | | | | |
| gemini-2.5-pro | 70 | 4.70 | 4.8 | 4.8 | 0 | 10 | 4.00 | 5.7 | 5.6 | 0 | 60 | 10.20 | 13.5 | 13.5 | 0 |
| gemini-2.5-flash | 0 | 0 | 0 | 0 | 0 | 0 | 0 | 0 | 0 | 0 | 0 | 1.90 | 4.5 | 1.6 | 0 |
| qwen-2.5-omni | 0 | 0.10 | 0.1 | 0.3 | 60 | 0 | 0 | 0 | 0 | 50 | 0 | 0 | 0 | 0.1 | 60 |
| MiniCPM-o-2.6 | 0 | 0.10 | 0.2 | 0.2 | 50 | 0 | 0.10 | 0.1 | 0.3 | 20 | 0 | 0 | 0 | 0 | 40 |
| *Simplified Task* | | | | | | | | | | | | | | | |
| gemini-2.5-pro | 90 | 5.60 | 5.6 | 5.6 | 0 | 60 | 6.00 | 6.0 | 6.0 | 0 | 70 | 10.60 | 10.6 | 10.6 | 0 |
| gemini-2.5-flash | 0 | 0 | 0 | 0 | 0 | 10 | 1.00 | 1.0 | 1.0 | 0 | 10 | 1.55 | 1.5 | 1.6 | 0 |
| qwen-2.5-omni | 0 | 0.25 | 0.1 | 0.4 | 50 | 0 | 0.20 | 0.2 | 0.2 | 50 | 0 | 0.05 | 0 | 0.1 | 70 |
| MiniCPM-o-2.6 | 0 | 0.10 | 0.1 | 0.1 | 30 | 0 | 0 | 0 | 0 | 10 | 0 | 0 | 0 | 0.4 | 10 |

- **Instruction-Tuning Data Distribution:** Proprietary models are likely trained on vastly larger and more diverse instruction-following datasets, specifically including "rule-application in context" scenarios, a capability directly tested here.

- **Architectural Differences:** There may be architectural differences in how textual context from prompts is integrated with real-time sensory inputs. Proprietary models might employ specialized attention mechanisms that maintain and query contextual instructions during inference, a feature potentially less developed in their open-source counterparts.

This gap has significant implications. While our core findings suggest models struggle with autonomous fusion, this experiment shows that even explicit scaffolding only helps the largest proprietary models. Promising research directions include curriculum learning for rule-based reasoning and architectural innovations like explicit "rule memory" modules.

## F.5 TASK SIMPLIFICATION

To validate the designed complexity of our benchmark, we conducted a task simplification experiment on *Myriad Echoes*. The goal was to understand if the primary difficulty lay in the multi-modal parsing phase or the long-sequence execution phase.

**Methodology.** We modified the original two-phase task into a single-phase perception task. In this simplified version, the agent still observes the full multi-modal sequence (video and audio) as in Phase 1. However, instead of proceeding to a second execution phase, the agent's task is to directly output the symbolic sequence it perceived. The performance is then measured by a final score which is a weighted average of the coordinate accuracy (50%) and the icon accuracy (50%) of its output.

**Results.** The results of this experiment are presented in Table 17, compared against the baseline performance on the original task. As expected, all models showed performance gains on the simplified task, as it removes the challenging long-horizon execution and action grounding components. However, even on this simplified perception-only task, the weaker open-source models still struggled to achieve high accuracy, particularly on the Hard difficulty. This confirms that the benchmark's core challenges are substantial and distributed across both its perception and action phases.

## F.6 MODALITY SUBSTITUTION

This final diagnostic experiment investigates the models' ability to generalize across different modality representations of the same semantic information. Specifically, we tested if agents perform better when complex information is presented as structured text versus synthesized audio.

**Methodology.** We used the *Phantom Soldiers in the Fog* environment (Medium difficulty) for this experiment. In the baseline condition, the agent receives tactical guidance via the audio channel (as Text-to-Speech). In the 'Modality Substitution' condition, we disabled the audio channel entirely. Instead, the exact same structured textual guidance that would have been converted to speech was appended directly to the main text prompt. The agent's task was then to complete the mission using only the visual (video) and augmented textual modalities.

**Results.** The results, presented in Table 18, reveal a strong and consistent trend. Most models, particularly the high-performing proprietary ones, showed a significant performance *increase* when the auditory information was substituted with its textual equivalent. For example, Gemini 2.5 Pro's score improved from 78.81 to 86.78, and Gemini 2.5 Flash's score more than doubled from 31.2 to 70.2. This unexpected result reinforces our core finding about brittle fusion for non-textual information. It suggests that for current models, well-structured and unambiguous text is a more reliable and easier-to-process source of information than synthesized audio, even when the underlying semantic content is identical. The performance drop for MiniCPM-o-2_6 is an anomaly that warrants further investigation, but may point to architectural differences in how it handles combined textual inputs.

Table 18: Performance comparison on *Phantom Soldiers in the Fog* (Medium) with and without Modality Substitution.

| Model | Baseline (Video+Text+Audio) | | Substituted (Video+Text only) | |
|---|---|---|---|---|
| | Score | Succ. Rate | Score | Succ. Rate |
| gemini-2.5-pro | 78.81 | 0.880 | 86.78 | 0.920 |
| gemini-2.5-flash | 31.20 | 0.570 | 70.20 | 0.860 |
| MiniCPM-o-2.6 | 11.60 | 0.200 | 1.80 | 0.035 |
| qwen-2.5-omni | 23.74 | 0.465 | 25.90 | 0.470 |

# G QUALITATIVE CASE STUDIES

To provide deeper, qualitative insights into the quantitative results presented in the main text, this section presents detailed case studies for three noteworthy phenomena observed during our evaluation.

## G.1 CASE STUDY: SUPERHUMAN MEMORY IN MYRIAD ECHOES

**Phenomenon.** As noted in the main text, top-tier models like Gemini 2.5 Pro exhibit superhuman performance on the *Myriad Echoes* task, particularly on Hard difficulty where the sequence length is long. This case study analyzes the cognitive and architectural differences between the AI agent and a human player that lead to this performance gap.

**Analysis.** The core challenge of *Myriad Echoes* is twofold: high-bandwidth, cross-modal information encoding followed by precise, long-sequence symbolic execution. As illustrated in Figure 27, the model is presented with a rapid, lengthy sequence of icon-sound pairs.

- **Model's Advantage:** A large omni-modal model like Gemini 2.5 Pro functions as a near-perfect information transducer. Its vast parameter space and attention mechanisms allow it to faithfully transcribe the high-throughput audio-visual stream into a precise internal symbolic representation with minimal loss. In the execution phase, it can recall and act upon this long sequence with near-perfect accuracy, as its 'working memory' is not biologically constrained.

- **Human's Limitation:** In contrast, a human player's performance is fundamentally limited by the capacity of their cognitive working memory (typically cited as 7±2 items). It is cognitively impossible for a human to perfectly memorize a rapid sequence of 10 or more arbitrary audio-visual pairs. Humans must resort to chunking or other heuristics, which are prone to error and forgetting, leading to a much lower performance ceiling.

**Conclusion:** This task highlights a domain where current AI excels: high-fidelity, short-term memory and precise symbolic manipulation. The observed superhuman performance is an expected outcome of the architectural differences between the model and the human brain, rather than an indication of superior general reasoning.

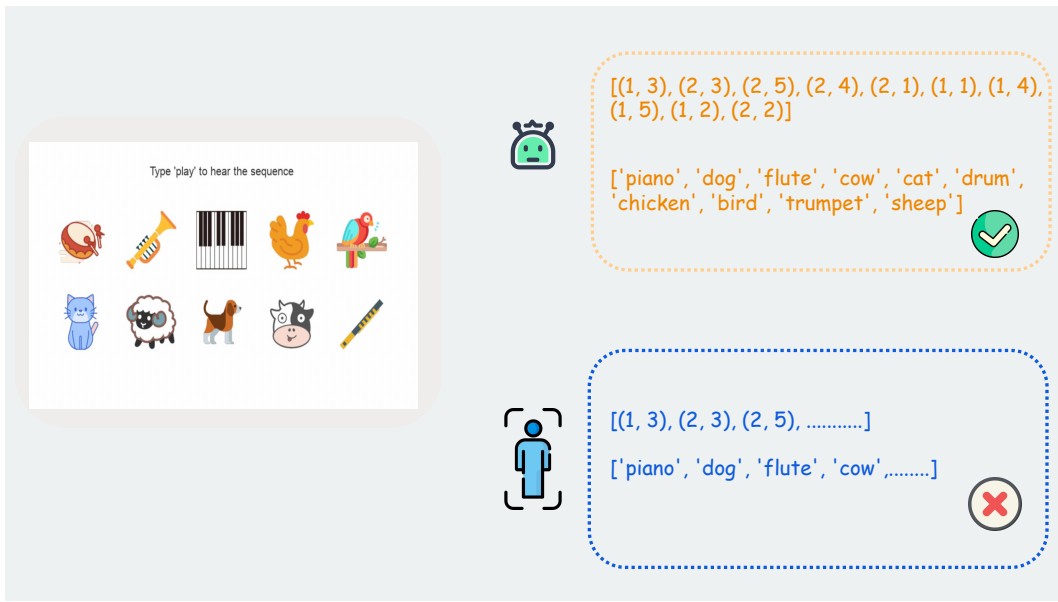

Figure 27: Illustration of the performance gap in *Myriad Echoes*. The AI agent (top) can perfectly recall and transcribe the long, complex audio-visual sequence. The human player (bottom) is limited by their working memory and cannot reliably recall the full sequence.

### G.2 CASE STUDY: ANOMALOUS WINNING STRATEGY IN BLASTING SHOWDOWN

**Phenomenon.** During the AI-vs-AI tournament in *Blasting Showdown*, we observed cases where MiniCPM-o-2_6 won matches despite having a Kill/Death (K/D) ratio of zero. This case study examines the unusual, passive strategy that led to this counter-intuitive success.

**Analysis.** The model's winning strategy can be characterized as extreme **risk aversion** or **passive survival**. Figure 28 depicts a representative match.

- **Observed Behavior:** Throughout the match, MiniCPM-o-2_6 (represented by the red player, Player 1) exhibited a very low tendency to place bombs or engage opponents. Its primary behavior consisted of reactive movements to evade bombs placed by other, more aggressive agents.

- **Environmental Dynamics:** The other three agents (Players 2, 3, and 4) actively engaged in combat. This created a chaotic and dangerous environment where players were eliminated not just by direct attacks, but also by chain reactions, self-elimination (getting trapped by their own bomb), or being caught in crossfire. In the depicted sequence, Player 2 eliminates Players 4 and 3, but then accidentally traps and eliminates itself.

- **Attribution:** It is unlikely that the model devised a sophisticated, deliberate strategy of 'waiting out the storm'. A more plausible explanation is that its capacity for proactive, strategic planning is underdeveloped, causing it to default to the simplest possible policy: stay alive by avoiding immediate threats. In the chaotic context of a 4-player free-for-all, this simple, passive policy coincidentally proved to be highly effective.

**Conclusion:** This case study is a crucial reminder that in complex multi-agent systems, a successful outcome does not necessarily imply intelligent strategy. It highlights the importance of analyzing an agent's behavioral traces, not just its win rate, to accurately assess its planning and reasoning capabilities.

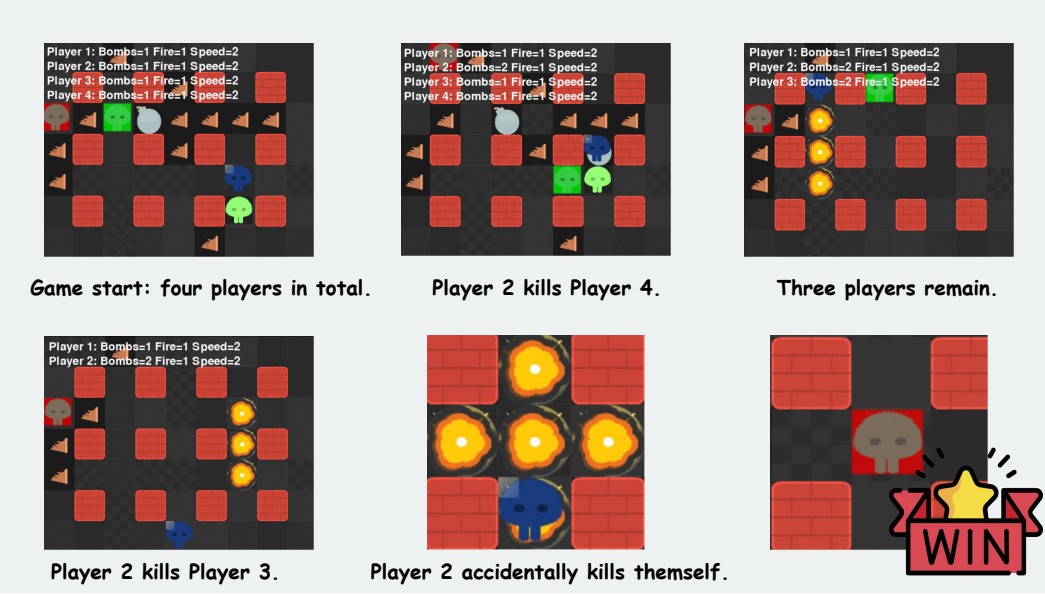

Figure 28: A step-by-step replay of a match won by MiniCPM-o-2_6 (Player 1, red). Player 1 remains passive while the other agents eliminate each other through aggressive play and miscalculation, leading to an accidental victory.

### G.3 CASE STUDY: SYSTEMATIC FAILURE IN MYRIAD ECHOES

**Phenomenon.** A peculiar and consistent failure mode was observed for Gemini 2.5 Flash in the *Myriad Echoes* task. On Easy and Medium difficulties, its performance on all sequence-related metrics was consistently zero.

**Analysis.** The root cause of this total failure is a systematic **off-by-one error** in its sequence generation. Figure 29 provides a clear example.

- **The Task:** The agent is presented with a true sequence of a specific length (e.g., 10 items in the example).
- **The Error:** When prompted to reproduce the sequence, Gemini 2.5 Flash consistently outputs a sequence that is correct in content and relative order, but is missing the first element. It always generates a sequence of length N-1 when the correct length is N.
- **Attribution:** This behavior points to a subtle but critical flaw in how the model handles sequence boundaries or follows length constraints. It is a classic example of a **format following failure**. The model understands the core task of identifying the items, but fails on the crucial meta-task of adhering to the sequence's structural integrity (in this case, its length).

**Conclusion:** This case demonstrates that even highly capable models can harbor specific, systematic bugs in their reasoning or generation processes. It highlights the value of diagnostic benchmarks like OmniPlay, which can surface these otherwise hidden, granular failure modes that would be missed by evaluations that only measure average performance. For tasks requiring high precision, such a systematic error is a critical failure.

## H DETAILED STATISTICAL RESULTS FOR DIAGNOSTIC EXPERIMENTS

This section provides the full statistical data for the diagnostic experiments presented in Section 5.2, including the modality conflict and modality ablation studies. All experiments were conducted over N=50 independent runs to ensure statistical robustness.

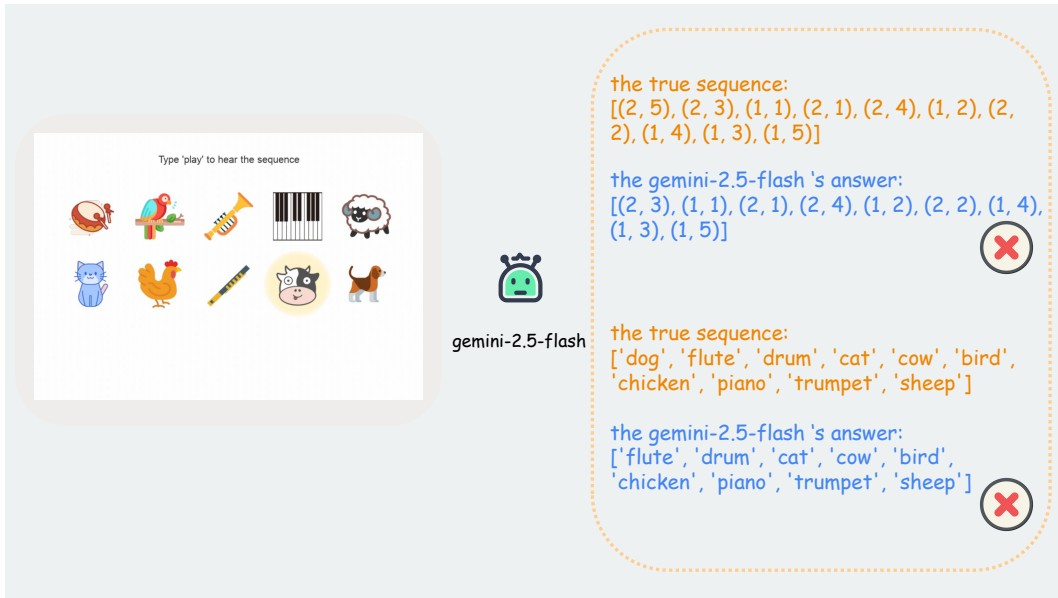

Figure 29: An example of Gemini 2.5 Flash's systematic 'off-by-one' error. The model correctly identifies most of the sequence but consistently omits the first element, resulting in a complete task failure.

## H.1 MODALITY CONFLICT (SUPPORTING FIGURE 4B)

Table 19 contains the detailed statistical results for the modality conflict experiment performed on *Whispered Pathfinding (Hard)*. These results are visualized in Figure 4b in the main text. We report the mean efficiency score, the standard deviation (SD) to show the raw performance volatility, and the standard error of the mean (SEM) used for generating the error margins in the figure.

Table 19: Full statistical results for the Modality Conflict experiment on *Whispered Pathfinding (Hard)*. The data corresponds to Figure 4b. All metrics are based on N=50 runs.

| Model | Condition | Mean Score (%) | SD | SEM |
|---|---|---|---|---|
| **Gemini 2.5 Pro** | No Conflict | 89.4 | 5.7 | 0.8 |
| | Audio Conflict | 43.3 | 16.3 | 2.3 |
| | Text Conflict | 32.2 | 17.7 | 2.5 |
| **Gemini 2.5 Flash** | No Conflict | 89.0 | 6.4 | 0.9 |
| | Audio Conflict | 88.6 | 7.1 | 1.0 |
| | Text Conflict | 48.1 | 20.5 | 2.9 |
| **Qwen-2.5-Omni (7B)** | No Conflict | 45.9 | 22.6 | 3.2 |
| | Audio Conflict | 0.0 | 0.0 | 0.0 |
| | Text Conflict | 37.6 | 24.0 | 3.4 |
| **MiniCPM-o-2.6 (8B)** | No Conflict | 59.5 | 19.1 | 2.7 |
| | Audio Conflict | 32.2 | 24.7 | 3.5 |
| | Text Conflict | 56.8 | 21.2 | 3.0 |

## H.2 MODALITY ABLATION (SUPPORTING FIGURE 5)

Table 20 provides the detailed statistical results for the modality ablation experiment, corresponding to Figure 5 in the main text. The experiment was conducted on the 'Hard' difficulty for two distinct tasks: *Whispered Pathfinding* and *Myriad Echoes*. All metrics are based on N=50 independent runs.

Table 20: Full statistical results for the Modality Ablation experiment. The data corresponds to Figure 5.

| Task | Model | Condition | Mean Score | SD | SEM |
|------|-------|-----------|-----------|-----|-----|
| *Whispered Pathfinding* (Efficiency Score, %) | **Gemini 2.5 Pro** | Full Modality | 86.7 | 8.5 | 1.2 |
| | | Removed Audio | 48.9 | 17.0 | 2.4 |
| | | Removed Image | 80.8 | 9.9 | 1.4 |
| | | Removed Text | 37.2 | 18.4 | 2.6 |
| | **Gemini 2.5 Flash** | Full Modality | 86.0 | 9.2 | 1.3 |
| | | Removed Audio | 79.4 | 12.0 | 1.7 |
| | | Removed Image | 55.5 | 17.7 | 2.5 |
| | | Removed Text | 83.5 | 10.6 | 1.5 |
| | **Qwen-2.5-Omni (7B)** | Full Modality | 31.6 | 21.2 | 3.0 |
| | | Removed Audio | 0.0 | 0.0 | 0.0 |
| | | Removed Image | 27.0 | 22.6 | 3.2 |
| | | Removed Text | 60.2 | 24.0 | 3.4 |
| | **MiniCPM-o-2.6 (8B)** | Full Modality | 48.8 | 25.5 | 3.6 |
| | | Removed Audio | 47.0 | 26.9 | 3.8 |
| | | Removed Image | 81.4 | 11.3 | 1.6 |
| | | Removed Text | 67.5 | 21.9 | 3.1 |
| *Myriad Echoes* (Weighted Score) | **Gemini 2.5 Pro** | Full Modality | 11.85 | 1.50 | 0.21 |
| | | Removed Audio | 9.80 | 2.12 | 0.30 |
| | | Removed Image | 9.80 | 2.12 | 0.30 |
| | | Removed Text | 8.00 | 2.47 | 0.35 |
| | **Gemini 2.5 Flash** | Full Modality | 2.48 | 3.18 | 0.45 |
| | | Removed Audio | 0.90 | 1.77 | 0.25 |
| | | Removed Image | 2.38 | 2.97 | 0.42 |
| | | Removed Text | 0.35 | 1.13 | 0.16 |
| | **Qwen-2.5-Omni (7B)** | Full Modality | 0.03 | 0.14 | 0.02 |
| | | Removed Audio | 0.18 | 0.42 | 0.06 |
| | | Removed Image | 0.13 | 0.35 | 0.05 |
| | | Removed Text | 0.00 | 0.00 | 0.00 |
| | **MiniCPM-o-2.6 (8B)** | Full Modality | 0.00 | 0.00 | 0.00 |
| | | Removed Audio | 0.03 | 0.14 | 0.02 |
| | | Removed Image | 0.08 | 0.28 | 0.04 |
| | | Removed Text | 0.40 | 0.99 | 0.14 |

## I   FULL PERFORMANCE RESULTS

This appendix provides the complete, unabridged performance data for all models and baselines across all tasks and difficulty levels from our main evaluation. We first present the summary statistics for all NPS-benchmarked tasks, which directly support the main findings in Section 5.1. Following this, we provide the detailed, task-specific raw metrics for each game environment.

### I.1   SUMMARY OF NPS-BENCHMARKED TASKS

Table 21 provides the complete statistical data corresponding to the results visualized in Figure 3a and summarized in Table **??**. For each model and task, we conducted 50 independent runs with different random seeds. We report the Mean Normalized Performance Score (NPS), the Standard Deviation (SD) to show performance volatility, and the Standard Error of the Mean (SEM) to indicate the confidence in our estimation of the mean.

Table 21: Full statistical results for all NPS-benchmarked tasks (N=50 runs). Data is presented as Mean NPS.

| Task (Grouped by Game) | Model | Mean NPS | Standard Deviation (SD) | Standard Error (SEM) |
|---|---|---|---|---|
| *Whispered Pathfinding* (Easy) | Gemini 2.5 Pro | 98.2 | 4.2 | 0.6 |
| | Gemini 2.5 Flash | 95.9 | 7.1 | 1.0 |
| | Qwen-2.5-Omni | 66.6 | 21.2 | 3.0 |
| | MiniCPM-o-2.6 | 86.8 | 15.6 | 2.2 |
| | Baichuan-Omni-1.5 | 88.0 | 14.1 | 2.0 |
| | VITA-1.5 | 78.6 | 17.7 | 2.5 |
| *Whispered Pathfinding* (Medium) | Gemini 2.5 Pro | 99.2 | 3.5 | 0.5 |
| | Gemini 2.5 Flash | 95.7 | 8.5 | 1.2 |
| | Qwen-2.5-Omni | 78.6 | 19.8 | 2.8 |
| | MiniCPM-o-2.6 | 90.8 | 12.0 | 1.7 |
| | Baichuan-Omni-1.5 | 93.2 | 9.9 | 1.4 |
| | VITA-1.5 | 84.4 | 16.3 | 2.3 |
| *Whispered Pathfinding* (Hard) | Gemini 2.5 Pro | 95.2 | 8.5 | 1.2 |
| | Gemini 2.5 Flash | 95.0 | 9.2 | 1.3 |
| | Qwen-2.5-Omni | 75.6 | 23.3 | 3.3 |
| | MiniCPM-o-2.6 | 81.7 | 18.4 | 2.6 |
| | Baichuan-Omni-1.5 | 85.0 | 16.3 | 2.3 |
| | VITA-1.5 | 82.7 | 19.1 | 2.7 |
| *Myriad Echoes* (Easy) | Gemini 2.5 Pro | 114.1 | 12.0 | 1.7 |
| | Gemini 2.5 Flash | -7.7 | 31.8 | 4.5 |
| | Qwen-2.5-Omni | -3.8 | 36.1 | 5.1 |
| | MiniCPM-o-2.6 | -3.8 | 38.2 | 5.4 |
| | Baichuan-Omni-1.5 | -4.5 | 37.5 | 5.3 |
| | VITA-1.5 | -6.1 | 41.0 | 5.8 |
| *Myriad Echoes* (Medium) | Gemini 2.5 Pro | 157.0 | 18.4 | 2.6 |
| | Gemini 2.5 Flash | -2.5 | 33.2 | 4.7 |
| | Qwen-2.5-Omni | -2.5 | 39.6 | 5.6 |
| | MiniCPM-o-2.6 | 2.5 | 43.1 | 6.1 |
| | Baichuan-Omni-1.5 | 0.0 | 40.3 | 5.7 |
| | VITA-1.5 | -2.3 | 44.5 | 6.3 |
| *Myriad Echoes* (Hard) | Gemini 2.5 Pro | **399.2** | 25.5 | 3.6 |
| | Gemini 2.5 Flash | 81.4 | 29.7 | 4.2 |
| | Qwen-2.5-Omni | -1.7 | 34.6 | 4.9 |
| | MiniCPM-o-2.6 | -2.5 | 38.9 | 5.5 |
| | Baichuan-Omni-1.5 | -2.5 | 37.5 | 5.3 |
| | VITA-1.5 | -2.5 | 41.7 | 5.9 |
| *The Alchemist's Melody* (Default) | Gemini 2.5 Pro | 28.4 | 33.2 | 4.7 |
| | Gemini 2.5 Flash | 10.5 | 38.9 | 5.5 |
| | Qwen-2.5-Omni | 9.2 | 41.0 | 5.8 |
| | MiniCPM-o-2.6 | 7.7 | 43.1 | 6.1 |
| | Baichuan-Omni-1.5 | 10.2 | 39.6 | 5.6 |
| | VITA-1.5 | -8.9 | 48.1 | 6.8 |
| *Phantom Soldiers* (Easy) | Gemini 2.5 Pro | 88.6 | 19.8 | 2.8 |
| | Gemini 2.5 Flash | 86.5 | 22.6 | 3.2 |
| | Qwen-2.5-Omni | -25.6 | 45.2 | 6.4 |
| | MiniCPM-o-2.6 | -28.4 | 48.1 | 6.8 |
| | Baichuan-Omni-1.5 | 16.5 | 38.9 | 5.5 |
| | VITA-1.5 | -38.1 | 53.0 | 7.5 |
| *Phantom Soldiers* (Medium) | Gemini 2.5 Pro | 73.6 | 26.9 | 3.8 |
| | Gemini 2.5 Flash | 6.3 | 36.8 | 5.2 |
| | Qwen-2.5-Omni | -9.1 | 43.1 | 6.1 |
| | MiniCPM-o-2.6 | -42.2 | 50.9 | 7.2 |
| | Baichuan-Omni-1.5 | -35.4 | 49.5 | 7.0 |
| | VITA-1.5 | -69.3 | 35.4 | 5.0 |
| *Phantom Soldiers* (Hard) | Gemini 2.5 Pro | **87.5** | 24.7 | 3.5 |
| | Gemini 2.5 Flash | 54.5 | 33.2 | 4.7 |
| | Qwen-2.5-Omni | 11.2 | 41.0 | 5.8 |
| | MiniCPM-o-2.6 | -21.5 | 49.5 | 7.0 |
| | Baichuan-Omni-1.5 | 8.3 | 43.8 | 6.2 |
| | VITA-1.5 | **-49.2** | 42.4 | 6.0 |

Table 22: Summary of Human Expert Performance (N=12 participants). Data is presented as the mean raw score used for NPS calculation, along with the standard deviation (SD) and standard error of the mean (SEM) measuring inter-player variability. Scoring rules are detailed in Appendix D.

| Task (Grouped by Game) | Mean Raw Score | SD (Inter-Player) | SEM (Inter-Player) |
|---|---|---|---|
| *Whispered Pathfinding* (Easy) | 0.20 | 0.03 | 0.009 |
| *Whispered Pathfinding* (Medium) | 0.13 | 0.02 | 0.006 |
| *Whispered Pathfinding* (Hard) | 0.07 | 0.01 | 0.003 |
| *Myriad Echoes* (Easy) | 4.20 | 0.38 | 0.11 |
| *Myriad Echoes* (Medium) | 3.10 | 0.28 | 0.08 |
| *Myriad Echoes* (Hard) | 3.03 | 0.36 | 0.10 |
| *The Alchemist's Melody* (Default) | 87.66 | 6.14 | 1.77 |
| *Phantom Soldiers* (Easy) | 100.00 | 4.50 | 1.30 |
| *Phantom Soldiers* (Medium) | 98.80 | 9.39 | 2.71 |
| *Phantom Soldiers* (Hard) | 96.75 | 12.58 | 3.63 |

## I.2 STATISTICAL ANALYSIS OF WIN RATES IN BLASTING SHOWDOWN

Table 23 provides the statistical analysis for the win rates reported in Figure 4. The analysis is based on the outcomes of a 50-game tournament. We report the raw number of wins, the mean win rate (p), the standard deviation (SD) calculated as $\sqrt{p(1-p)}$, and the standard error of the mean (SEM) calculated as $SD/\sqrt{N}$, where N=50.

Table 23: Statistical analysis of win rates for the AI-vs-AI evaluation on *Blasting Showdown* (N=50 games).

| Model | Wins / Total | Win Rate (%) | SD | SEM (%) |
|---|---|---|---|---|
| Gemini 2.5 Pro | 18 / 50 | 36.1% | 0.480 | 6.8% |
| Gemini 2.5 Flash | 14 / 50 | 28.9% | 0.453 | 6.4% |
| MiniCPM-o-2.6 | 10 / 50 | 19.4% | 0.395 | 5.6% |
| Baichuan-Omni-1.5 | 9 / 50 | 17.7% | 0.382 | 5.4% |
| Qwen-2.5-Omni | 6 / 50 | 11.8% | 0.323 | 4.6% |
| VITA-1.5 | 4 / 50 | 7.4% | 0.262 | 3.7% |

## I.3 TASK-SPECIFIC RAW METRICS: WHISPERED PATHFINDING

Table 24 presents the detailed performance metrics for the *Whispered Pathfinding* task. The primary metric for this navigation task is 'Mean Steps', where a lower value indicates better performance. We also report the trimmed mean, which is less sensitive to outliers.

Table 24: Full performance results for *Whispered Pathfinding* across all difficulties.

| Model | Easy Difficulty | | | | | Medium Difficulty | | | | | Hard Difficulty | | | | |
|---|---|---|---|---|---|---|---|---|---|---|---|---|---|---|---|
| | Mean | Min | Max | Inv. | Trim. | Mean | Min | Max | Inv. | Trim. | Mean | Min | Max | Inv. | Trim. |
| human | 5.2 | 3 | 8 | 0.0 | 5.1 | 8.3 | 6 | 10 | 0.0 | 8.0 | 15.6 | 10 | 27 | 0.0 | 13.9 |
| gemini-2.5-pro | 7.6 | 5 | 10 | 0.0 | 7.6 | 10.2 | 7 | 14 | 0.0 | 10.1 | 42.6 | 13 | 152 | 0.0 | 36.2 |
| gemini-2.5-flash | 16.1 | 4 | 70 | 0.8 | 10.9 | 23.2 | 6 | 87 | 1.2 | 19.0 | 43.5 | 18 | 112 | 2.7 | 37.15 |
| qwen-2.5-omni | 70.3 | 10 | 273 | 29.5 | 52.5 | 64.2 | 11 | 132 | 28.1 | 62.4 | 130.1 | 32 | 253 | 52.6 | 128.2 |
| MiniCPM-o-2.6 | 27.0 | 7 | 73 | 7.5 | 23.8 | 34.4 | 6 | 86 | 10.7 | 31.5 | 110.8 | 34 | 255 | 35.8 | 99.5 |
| VITA-1.5 | 36.1 | 13 | 70 | 15.0 | 35.5 | 52.8 | 8 | 162 | 21.9 | 47.8 | 106.5 | 23 | 343 | 35.7 | 94.8 |
| Baichuan-Omni-1.5 | 23.1 | 11 | 47 | 6.3 | 22.2 | 31.0 | 12 | 67 | 7.8 | 25.3 | 89.3 | 27 | 236 | 20.2 | 84.7 |
| random | 193.2 | 25 | 500 | 0.0 | 147.0 | 277.8 | 125 | 477 | 0.0 | 262.3 | 413.4 | 119 | 500 | 0.0 | 482.7 |

## I.4 TASK-SPECIFIC RAW METRICS: MYRIAD ECHOES

Table 25 presents the detailed performance metrics for the *Myriad Echoes* task. This task assesses both multi-modal parsing (Coord. Acc., Icon Acc.) and execution (Mean Score).

Table 25: Full performance results for *Myriad Echoes* across all difficulties.

| Model | Easy Difficulty | | | | | Medium Difficulty | | | | | Hard Difficulty | | | | |
|---|---|---|---|---|---|---|---|---|---|---|---|---|---|---|---|
| | Succ(%) | M.Score | Coord. | Icon | ParseF(%) | Succ(%) | M.Score | Coord. | Icon | ParseF(%) | Succ(%) | M.Score | Coord. | Icon | ParseF(%) |
| human | - | 3.70 | 3.60 | 5.80 | - | - | 2.50 | 2.60 | 4.80 | - | - | 2.60 | 2.30 | 4.60 | - |
| gemini-2.5-pro | 70 | 4.70 | 4.80 | 4.80 | 0 | 10 | 4.00 | 5.70 | 5.60 | 0 | 60 | 10.20 | 13.50 | 13.50 | 0 |
| gemini-2.5-flash | 0 | 0 | 0 | 0 | 0 | 0 | 0 | 0 | 0 | 0 | 0 | 1.90 | 4.50 | 1.60 | 0 |
| qwen-2.5-omni | 0 | 0.10 | 0.10 | 0.30 | 60 | 0 | 0 | 0 | 0 | 50 | 0 | 0 | 0 | 0.10 | 60 |
| MiniCPM-o-2.6 | 0 | 0.10 | 0.20 | 0.20 | 50 | 0 | 0.10 | 0.10 | 0.30 | 20 | 0 | 0 | 0 | 0 | 40 |
| VITA-1.5 | 0 | 0.10 | 0.05 | 0 | 0 | 0 | 0 | 0 | 0.02 | 20 | 0 | 0 | 0 | 0 | 30 |
| Baichuan-Omni-1.5 | 0 | 0.20 | 0.10 | 0 | 50 | 0 | 0.10 | 0.10 | 0 | 70 | 0 | 0 | 0 | 0 | 60 |
| random | 0 | 0.55 | 0.08 | 0.02 | 0 | 0 | 0.05 | 0.05 | 0.15 | 0 | 0 | 0.10 | 0.07 | 0.03 | 0 |

## I.5 TASK-SPECIFIC RAW METRICS: THE ALCHEMIST'S MELODY

Table 26 presents the detailed performance metrics for the *The Alchemist's Melody* task.

Table 26: Full performance results for *The Alchemist's Melody*.

| Model | Score | Completion Rate (%) |
|---|---|---|
| human | 87.66 | 100% |
| gemini-2.5-pro | 43.15 | 20% |
| gemini-2.5-flash | 32.05 | 0% |
| qwen-2.5-omni | 31.23 | 0% |
| MiniCPM-o-2.6 | 30.29 | 0% |
| VITA-1.5 | 20.01 | 0% |
| Baichuan-Omni-1.5 | 31.82 | 0% |
| random | 25.51 | 0% |

## I.6 TASK-SPECIFIC RAW METRICS: PHANTOM SOLDIERS IN THE FOG

Table 27 presents the detailed performance metrics for the *Phantom Soldiers in the Fog* task across all difficulties.

Table 27: Full performance results for *Phantom Soldiers in the Fog* across all difficulties.

| Model | Score | | | Success Rate | | |
|---|---|---|---|---|---|---|
| | Easy | Medium | Hard | Easy | Medium | Hard |
| human | 100.0 | 99.60 | 98.50 | 1.00 | 0.98 | 0.950 |
| gemini-2.5-pro | 83.51 | 78.81 | 91.62 | 1.00 | 0.88 | 0.857 |
| gemini-2.5-flash | 80.39 | 31.20 | 73.54 | 1.00 | 0.57 | 0.610 |
| qwen-2.5-omni | 5.13 | 23.74 | 23.34 | 0.13 | 0.465 | 0.550 |
| MiniCPM-o-2.6 | 3.10 | 11.60 | 8.93 | 0.11 | 0.20 | 0.270 |
| VITA-1.5 | 0 | 0 | 0 | 0 | 0 | 0 |
| Baichuan-Omni-1.5 | 29.15 | 11.50 | 19.60 | 0.50 | 0.28 | 0.550 |
| random | 25.20 | 22.86 | 17.80 | 0.30 | 0.58 | 0.460 |

## I.7 TASK-SPECIFIC RAW METRICS: BLASTING SHOWDOWN

Table 28 presents the full tournament results for the *Blasting Showdown* task. As this is a competitive multi-agent environment, performance is measured by win rates and combat effectiveness metrics rather than a normalized score.

## J IN-CONTEXT LEARNING EXPERIMENT

To distinguish between a fundamental model incapacity and a lack of task-specific adaptation (as queried by Reviewer 3), we conducted a comprehensive 3-shot In-Context Learning (ICL) experiment on *Phantom Soldiers in the Fog* across all three difficulty levels (Easy, Medium, Hard).

Table 28: Full tournament results for the AI-vs-AI evaluation on *Blasting Showdown*.

| Model | Games Played | Wins | Win Rate (%) | Kills | Deaths | K/D Ratio |
|---|---|---|---|---|---|---|
| gemini-2.5-pro | 36 | 13 | 36.11% | 93 | 39 | 2.38 |
| gemini-2.5-flash | 38 | 11 | 28.95% | 68 | 41 | 1.66 |
| MiniCPM-o-2.6 | 31 | 6 | 19.35% | 0 | 72 | 0.00 |
| Baichuan-Omni-1.5 | 34 | 6 | 17.65% | 31 | 55 | 0.56 |
| qwen-2.5-omni | 34 | 4 | 11.76% | 13 | 53 | 0.25 |
| VITA-1.5 | 27 | 2 | 7.41% | 0 | 42 | 0.00 |

## J.1 EXPERIMENTAL SETUP

We evaluated **Gemini 2.5 Pro** using a prompt augmented with:

1. **Strategic Guidelines:** Explicit rules derived from expert strategies (e.g., prioritization of audio cues).
2. **Expert Demonstrations:** Three curated examples of expert gameplay showing state analysis, reasoning, and decision-making.
3. **Chain-of-Thought Requirement:** The model was instructed to output an "Expert Analysis" before generating its final command to encourage reasoning.

To ensure statistical robustness, all results reported below are averaged over **20 independent episodes** per condition.

## J.2 RESULTS AND ANALYSIS

We compared the ICL performance against the Zero-shot baseline under both standard conditions and audio-visual conflict. The results are summarized in Table 29.

Table 29: Comparison of Zero-shot vs. 3-shot ICL performance for Gemini 2.5 Pro on *Phantom Soldiers* across varying difficulties. (Averaged over 20 episodes).

| Difficulty | Condition | Standard Score | Conflict Score | % Degradation |
|---|---|---|---|---|
| **Easy** | Zero-shot | 96.5 | 62.0 | -35.8% |
| | **3-shot ICL** | **98.1** | **64.5** | **-34.2%** |
| **Medium** | Zero-shot | 94.0 | 51.5 | -45.2% |
| | **3-shot ICL** | **97.5** | **54.0** | **-44.6%** |
| **Hard** | Zero-shot | 91.6 | 43.3 | -52.7% |
| | **3-shot ICL** | **96.7** | **47.2** | **-51.2%** |

**1. Adaptation is Successful Across All Levels:** ICL consistently improves performance in the standard setting. Notably, in the Hard difficulty, ICL boosts the score from 91.6 to **96.7**, effectively closing the gap with the human expert (98.5). This proves the model is capable of understanding the task and executing complex strategies when provided with context.

**2. Brittleness is Structural and Persistent:** Despite this successful adaptation, the model's robustness to modality conflict remains virtually unchanged.

- In **Hard** difficulty, the zero-shot model degrades by 52.7%, and the ICL model degrades by **51.2%**.
- This pattern holds across Easy and Medium difficulties, where the degradation percentages between Zero-shot and ICL are nearly identical (within ∼1%).

**Conclusion:** ICL successfully patches the "strategy gap" (raising the performance ceiling) but fails to remedy the "fusion gap" (the performance collapse under conflict remains severe). The consis-

tency of this degradation pattern across all difficulties strongly supports our hypothesis: the observed brittleness is a **fundamental architectural limitation** in processing conflicting sensory streams, rather than an artifact of zero-shot generalization.

## J.3 PROMPT STRUCTURE

The full prompt structure used for this experiment is provided below:

---

**Full ICL Prompt Structure**

**[System Prompt]** You are commanding a military team in a cooperative mission. You MUST provide EXACTLY ONE command per turn. ... *[Standard game mechanics: Hidden Objectives, Movement Uncertainty, Command Reliability]* ...

---

**\*\*\* STRATEGIC CONSIDERATIONS & GUIDELINES \*\*\***
- **Defense:** Protect High-Value Units. Medics and Scouts are critical; prioritize their survival over quick objective completion.
- **Offense:** Divide and Conquer. Split forces to cover more ground and encircle multiple objectives.
- **Conflict Resolution:** If Visual and Audio information conflict, prioritize **Audio alerts** and Hidden Hints as they are often more reliable in the fog.

---

**\*\*\* IN-CONTEXT LEARNING EXAMPLES \*\*\***
**Example 1: Defense & Survival - Protecting High-Value Units** *Situation:* Round 5/100. Medic (Member 2) has low health (55%) but is a high-value unit. Audio warns: "Medic reports: Need protection." *Expert Decision:* `COMMAND: 1 defend 45 48` *Reasoning:* "Use Heavy (Member 1) as a defensive barrier. Account for movement error when positioning. Protecting critical units takes priority over quick objective completion."
**Example 2: Offense & Coordination - Dividing Forces** *Situation:* Round 15/100. Two visible objectives are far apart. Audio indicates: "Multiple objectives detected." *Expert Decision:* `COMMAND: 0 recon 20 20` *Reasoning:* "Scout prioritizes exploring the first objective area (20, 20) due to high discovery rate. Splitting forces allows other members to handle the second objective simultaneously."
**Example 3: Conflict Resolution & Information Integration** *Situation:* Visual map shows area (65, 45) is safe. However, Audio report says: "Potential threat detected at (65, 45)." *Expert Analysis:* Information conflict detected (Audio vs. Visual). Scout source has high reliability. *Expert Decision:* `COMMAND: 0 recon 65 45` *Reasoning:* "Prioritized trusting the high-reliability information source (Scout Audio Report). Visual information may be incomplete due to fog/hidden units. Investigating the audio lead is safer than walking into a potential trap."

---

**\*\*\* COMMAND FORMAT \*\*\*** Provide EXACTLY ONE of the following:
- Individual: `COMMAND: [id] [action] [x] [y]`
- Team: `COMMAND: all [action] [x] [y]`

**\*\*\* NOW IT IS YOUR TURN \*\*\*** Current Game State: [Inserted dynamically]
**FINAL REMINDER:** Analyze the situation thoroughly. Choose the SINGLE most important action. Provide your strategic analysis, then end with exactly ONE command.

---

