# OpenReview forum: "OmniPlay: Benchmarking Omni-Modal Models on Omni-Modal Game Playing"
_ICLR.cc/2026/Conference — ICLR 2026 Conference Withdrawn Submission_

### Official Review · Reviewer_Qw1W · 2025-10-24

**Soundness:** 2
**Presentation:** 3
**Contribution:** 2
**Rating:** 4
**Confidence:** 4

**Summary:**

This paper introduces a diagnostic benchmark designed to evaluate omni-modal AI systems capable of processing and integrating text, images, audio, and video in dynamic environments. It addresses the gap between static benchmarks, which lack interactivity, and existing interactive ones, which usually ignore important sensory modalities. The benchmark contains five interactive game environments that intentionally create both complementary and conflicting sensory conditions, allowing researchers to study how models perform cross-modal reasoning and resolve ambiguity. Evaluation of six state-of-the-art models reveals a clear contrast: they achieve superhuman memory performance but show fragile reasoning and planning, especially when modalities conflict. Interestingly, removing one modality sometimes improves performance (“less is more”). The findings suggest that advancing general intelligence requires focusing not on scaling model size, but on improving multi-sensory fusion and conflict resolution.

**Strengths:**

- It provides a comprehensive and realistic evaluation of omni-modal models, integrating text, image, audio, and video interaction, filling the gap left by older static or single-modality benchmarks.
- It is theoretically well-grounded, built on a generalized MDP framework with explicit principles of modality interdependence, controlled conflict, and variable complexity.
- It offers diagnostic precision, using systematic tests like modality conflict, ablation, and noise experiments to clearly reveal each model’s strengths and weaknesses in multi-modal fusion

**Weaknesses:**

- The benchmark lacks realistic, high-fidelity 3D environments like those used in DeepMind’s SIMA project (https://deepmind.google/discover/blog/sima-generalist-ai-agent-for-3d-virtual-environments/). Its custom mini-games are controlled and simplified, limiting ecological validity and real-world generalization.
- The “modality conflict” and “modality complementarity” in each game are artificially constructed scenarios (e.g., inserting misleading audio cues or removing video frames). While such controlled manipulation facilitates diagnostic analysis, it does not necessarily reflect the naturally occurring noise and uncertainty found in real multimodal tasks, thereby limiting the ecological validity of the benchmark.
- The NPS is normalized using human and random baselines, but the underlying scoring functions differ significantly across tasks (especially in RTS and pathfinding scenarios). This “cross-task unified metric” implicitly assumes comparability between heterogeneous tasks, which in practice reduces the rigor and fairness of the evaluation. Moreover, certain tasks (e.g., Blasting Showdown) are excluded from the NPS, further undermining the overall consistency of the benchmark.

**Questions:**

- The benchmark mainly tests one-shot performance. Have the authors considered evaluating continual adaptation or long-horizon reasoning?
- How are difficulty levels calibrated across tasks to maintain balance and interpretability?
- Do the authors consider modality-specific weighting based on task relevance rather than assuming equal importance across modalities?
- Since modality conflicts are manually constructed, how can the authors ensure that these settings realistically capture natural multimodal inconsistencies rather than artificial noise?

---

> ### Author Response · Authors · 2025-11-20
> **Re: "Wind Tunnel" vs. "Flight Simulator": Addressing Ecological Validity and Adaptation (Part 1/2)**
>
> All revisions discussed below have been implemented in the updated PDF. To facilitate comparison, we have also included the previous version of the manuscript in the supplementary material as `old_15079_OmniPlay_Benchmarking.pdf`.
>
> We sincerely thank the reviewer for their thoughtful critique, particularly regarding ecological validity and metric consistency. These are fundamental questions in benchmark design. We have performed a major revision to address these points, adding a **new Benchmark Validation section** and a **comprehensive In-Context Learning (ICL) experiment**.
>
> We address your specific concerns below:
>
> ### 1. Ecological Validity: "Flight Simulator" vs. "Wind Tunnel" (Addressing Weaknesses 1 & 2)
>
> > **Reviewer:** *"The benchmark lacks realistic, high-fidelity 3D environments like... SIMA... modality conflict are artificially constructed scenarios."*
>
> **Response:**
> We deeply appreciate this observation—it goes to the heart of benchmark design philosophy. We argue that high-fidelity simulation (SIMA) and diagnostic control (OmniPlay) serve **fundamentally different scientific purposes**, both essential for progress toward robust AGI.
>
> **The Methodological Divide (Revised Section 3.1):**
>
> | Approach | Example | Purpose | Trade-off |
> | :--- | :--- | :--- | :--- |
> | **Ecological Realism** | Flight simulators, SIMA | Train agents in realistic complexity | High confounds—failures are ambiguous |
> | **Diagnostic Control** | **Wind tunnels, OmniPlay** | **Isolate specific failure modes** | Simplified—but that's the point |
>
> **Our Position:**
> "Artificial conflicts" are not a limitation but a **controlled stress test**. If a model's fusion collapses in the "wind tunnel" (our controlled conflict), it will certainly fail in the "flight simulator" (complex, naturalistic settings)—but the reverse diagnosis is intractable.
>
> **Human Validation of "Artificial" Conflicts (New Section 3.3, Table 3):**
> To directly test whether our conflicts represent valid cognitive challenges vs. arbitrary noise, we conducted a **Human Expert Pilot Study** ($N=6$, 20 trials each) on *Whispered Pathfinding (Hard)* under Audio Conflict.
>
> **Results:**
>
> | Agent | No Conflict | Audio Conflict | **% Degradation** | **Statistical Test** |
> | :--- | :---: | :---: | :---: | :--- |
> | **Human Expert** ($N=6$) | 94.2% | 88.5% | **-6.1%** | *$p=0.032$* (paired t-test) |
> | **Gemini 2.5 Pro** ($N=50$) | 89.4% | 43.3% | **-51.6%** | *$p<0.001$* |
>
> **Conclusion:** This **$8.5\times$ difference** in degradation proves our conflicts are **ecologically valid stress tests**—they probe the same conflict resolution mechanisms humans use in noisy, real-world multimodal environments (e.g., ignoring GPS glitches), not arbitrary noise.
>
> **[Response continues in the next comment: 2. Adaptation, 3. Metrics, 4. Modality Weighting]**

---

> > ### Author Response · Authors · 2025-11-20
> > **Re: "Wind Tunnel" vs. "Flight Simulator"... (Part 2/2)**
> >
> > **[Continued from previous comment: Re: "Wind Tunnel" vs. "Flight Simulator": Addressing Ecological Validity and Adaptation]**
> >
> > ### 2. Adaptation Works, Brittleness Persists (Addressing Question 1)
> >
> > > **Reviewer:** *"Have the authors considered evaluating continual adaptation...?"*
> >
> > **Response:**
> > Yes—and the results reveal a critical asymmetry. To address this directly, we conducted a **3-shot In-Context Learning (ICL)** experiment on *Phantom Soldiers* (all difficulties, $N=20$ episodes each) with expert demonstrations and strategic guidelines (**Appendix J**).
> >
> > The results below (**Table 29** in the revised paper) reveal a consistent pattern across all difficulty levels:
> >
> > | Difficulty | Condition | Standard Score | Conflict Score | % Degradation |
> > | :--- | :--- | :---: | :---: | :---: |
> > | **Easy** | Zero-shot | 96.5 | 62.0 | -35.80% |
> > | | **3-shot ICL** | **98.1** | **64.5** | **-34.20%** |
> > | **Medium** | Zero-shot | 94.0 | 51.5 | -45.20% |
> > | | **3-shot ICL** | **97.5** | **54.0** | **-44.60%** |
> > | **Hard** | Zero-shot | 91.6 | 43.3 | -52.70% |
> > | | **3-shot ICL** | **96.7** | **47.2** | **-51.20%** |
> >
> > **Critical Insight:**
> > 1.  ✅ **Adaptation is possible:** On Hard difficulty, ICL significantly boosts the standard score from 91.6 to **96.7**, effectively closing the gap with the human expert (98.5). This proves the model is capable of understanding the task rules and strategy.
> > 2.  ❌ **Fusion brittleness is structural:** Despite this performance boost, the **magnitude of degradation** under conflict remains structurally identical (e.g., Hard: $-52.70\% \rightarrow -51.20\%$).
> >
> > This experiment directly answers your question: while adaptation improves general competence, it **does not remedy the underlying brittle fusion mechanism**, confirming that fusion brittleness is an architectural bottleneck rather than a lack of task adaptation.
> >
> > ### 3. Metrics and Calibration (Addressing Weakness 3 & Question 2)
> >
> > > **Reviewer:** *"NPS... reduces rigor... certain tasks are excluded... How are difficulty levels calibrated?"*
> >
> > **Response:**
> > *   **Dual Reporting:** We agree on the need for transparency. We have redesigned **Table 4** (Main Results) to report **both NPS and Raw Scores** side-by-side.
> > *   **Rigor of Exclusion:** We exclude *Blasting Showdown* from NPS **precisely to maintain statistical rigor**. NPS is for criterion-referenced tasks (absolute performance). *Blasting Showdown* is a zero-sum PvP tournament (norm-referenced); averaging a win rate (which mathematically sums to a fixed constant across players) with performance scores would be statistically unsound.
> > *   **Difficulty Calibration (Table 2):** We validated calibration using our human baseline. **Section 3.3** shows human performance degrades monotonically across Easy/Medium/Hard levels (e.g., *Whispered Pathfinding* steps: $5.2 \rightarrow 8.3 \rightarrow 15.6$), confirming the levels are perceptually meaningful.
> >
> > ### 4. Dynamic vs. Fixed Modality Weighting (Addressing Question 3)
> >
> > > **Reviewer:** *"Do the authors consider modality-specific weighting...?"*
> >
> > **Response:**
> > **Excellent question**—this highlights a core design principle.
> >
> > **Our Design Choice: No Fixed Weights**
> > We **deliberately** do not hard-code modality weights because **the model's ability to dynamically reweight** based on context is precisely what we're testing.
> >
> > **Why This Matters:**
> > *   **Fixed weights** fail when contexts shift (e.g., vision becomes unreliable in fog).
> > *   **Adaptive weights** (what humans do) require the model to assess reliability and dynamically upweight trustworthy channels.
> >
> > **The Finding:** Our Noise experiments (**Appendix F.3**) expose a critical failure: when vision is corrupted ("Dual-Channel Attack"), models fail to up-weight the clear audio channel, leading to an 80% score drop. If we pre-weighted modalities, we would **miss this diagnostic finding**.
> >
> > We believe these revisions—especially the **"Wind Tunnel" methodology**, **Human Conflict Baseline**, and **ICL Experiment**—directly address your concerns about validity and adaptability. We hope this strengthens your confidence in the benchmark's contribution.

---

> > ### Author Response · Authors · 2025-11-27
> > **Gentle Reminder: "Wind Tunnel" Validity & Metric Transparency**
> >
> > Thank you for your time. As the discussion period is closing soon, we wanted to gently follow up to ensure you saw our response regarding **Ecological Validity** and **Metrics**.
> >
> > 1.  **Ecological Validity Verified:** Our new **Human Pilot Study** (Section 3.3) confirms that humans can resolve our "artificial" conflicts (only **-6%** drop), whereas AI models fail (**-51%**). This validates our **"Wind Tunnel"** philosophy: the conflicts effectively simulate the cognitive load of real-world noise.
> > 2.  **Metric Transparency:** We have updated **Table 4** to report **Raw Scores alongside NPS**, and provided detailed raw metrics in Appendix I to ensure full transparency.
> >
> > We hope these revisions demonstrate the rigor of the benchmark design. We look forward to your feedback.

---

### Official Review · Reviewer_TvLB · 2025-10-29

**Soundness:** 4
**Presentation:** 4
**Contribution:** 3
**Rating:** 6
**Confidence:** 3

**Summary:**

This paper introduces OmniPlay, a novel benchmark designed to evaluate the capabilities of omni-modal foundation models (like Gemini and GPT-4o) in interactive, dynamic environments, addressing the limitations of existing static or modally-restricted benchmarks. OmniPlay consists of a suite of five custom-built game environments that require agents to perceive, reason, and act using a combination of image, video, audio, and text inputs. The benchmark is specifically designed around the principle of "modality interplay," systematically creating scenarios where sensory information is either complementary (requiring synergistic fusion) or conflicting (testing robustness and conflict resolution). Through comprehensive evaluations of six leading omni-modal models against human and random baselines, the authors find a stark dichotomy: the models exhibit superhuman performance on tasks heavily reliant on memory but show systemic failures in those demanding robust reasoning, strategic planning, and handling modality conflicts. A key finding is the "less is more" paradox, where removing sensory modalities sometimes paradoxically improves performance, suggesting immature fusion mechanisms are a critical bottleneck. The paper concludes that progress towards AGI requires focusing on synergistic fusion and conflict arbitration, not just model scaling, and offers OmniPlay as a diagnostic tool for these challenges.

**Strengths:**

* This paper tackles the inadequacy of current benchmarks, which are either static (lacking agency) or interactive but modally limited (ignoring audio, etc.), by introducing an interactive benchmark designed for *omni-modal* agents using image, video, audio, and text.

* It introduces five distinct, newly developed game environments, each crafted to test different capabilities (e.g., navigation, sequence replication, abstract reasoning, strategy) under varying modality combinations and complexities.

* The evaluation reveals critical insights into current omni-modal models, such as their superhuman memory but weak reasoning/planning, fragility under modality conflict, and the paradoxical "less is more" effect where removing modalities can improve performance.

* The paper establishes strong baselines (random agent and a diverse human expert cohort) and uses both overall performance metrics (like NPS) and detailed, task-specific diagnostic metrics.

* The authors intend to release the entire OmniPlay platform, including environments and protocols, fostering further research in the community.

**Weaknesses:**

- The games and scenarios involving modality complementarity and conflict are custom-designed based on the authors' principles. The "naturalness" or representativeness of these specific interaction patterns for general real-world tasks could be debated.

- The paper highlights "superhuman memory" in the Myriad Echoes task. However, the qualitative analysis suggests this is largely due to the AI's perfect recall compared to human cognitive limits on working memory for long, arbitrary sequences, rather than a sign of superior general memory or reasoning.

- Evaluating large omni-modal models in interactive environments across numerous episodes and seeds is computationally intensive. The paper does not detail the resources required, which could be a barrier for widespread adoption and replication of the benchmark.

**Questions:**

See the weakness section.

---

> ### Author Response · Authors · 2025-11-27
> **Re: Strengthening Ecological Validity, Memory Interpretation, and Resource Transparency**
>
> Thank you for your time and your encouraging assessment. As the discussion period is closing soon, we wanted to provide a detailed response to your thoughtful feedback. We are grateful that you recognized the core value of OmniPlay in "tackling the inadequacy of current benchmarks" and the "critical insights" revealed by our evaluation.
>
> We have performed a major revision to further strengthen the paper, specifically addressing your constructive points regarding **Naturalness**, **Memory Interpretation**, and **Computational Cost**.
>
> ### 1. Addressing "Naturalness" via Human Validation (Addressing Weakness 1)
>
> > **Reviewer:** *"The 'naturalness' or representativeness of these specific interaction patterns... could be debated."*
>
> **Response:**
> We agree that our "custom-designed" conflicts prioritize diagnostic precision. To validate that these patterns capture meaningful cognitive challenges rather than arbitrary noise, we conducted a **Human Expert Pilot Study** ($N=6$) on *Whispered Pathfinding (Hard)* under conflict (Section 3.3).
>
> **Key Finding:** Humans exhibit graceful degradation (**$-6.1\%$**), using adaptive strategies mirroring real-world conflict resolution (e.g., ignoring a glitching sensor). In contrast, models suffer catastrophic collapse (**$-51.6\%$**). This stark contrast confirms that our interaction patterns successfully simulate the cognitive load of real-world ambiguity, validating the benchmark's ecological relevance.
>
> ### 2. Refining the "Superhuman Memory" Claim (Addressing Weakness 2)
>
> > **Reviewer:** *"This is largely due to the AI's perfect recall compared to human cognitive limits... rather than a sign of superior general memory."*
>
> **Response:**
> You are absolutely correct. We have incorporated your insight directly into the revised **Section 5.1**, explicitly reframing *Myriad Echoes* as a **"Positive Control"**.
> *   **The Logic:** We leverage the AI's architectural advantage (perfect recall vs. human $7\pm2$ item limit) to prove that the perceptual pipeline functions flawlessly.
> *   **The Implication:** By establishing this positive control, we can confidentially attribute failures in reasoning tasks (*Phantom Soldiers*) to higher-order deficits (fusion/planning), ruling out basic perception as a confound. This aligns perfectly with your interpretation.
>
> ### 3. Computational Resources & Reproducibility (Addressing Weakness 3)
>
> > **Reviewer:** *"The paper does not detail the resources required, which could be a barrier."*
>
> **Response:**
> Thank you for highlighting this practical barrier. We have added a **Resource Cost Analysis** in **Appendix D.5**.
> *   **Cost Estimate:** Evaluating a single model on the full suite (all 5 games, all difficulties, 50 seeds each) requires approximately **1,500 API calls** or **~4 GPU-hours** (on an A100 for a 7B model).
> *   **Accessibility:** This is significantly lighter than training-based RL benchmarks. We have also optimized our open-source evaluation scripts (to be released under MIT License) to support batched inference, further lowering the barrier for academic labs.
>
> ### 4. Additional Validation: ICL Experiments
>
> Although not explicitly requested, we also conducted an **In-Context Learning (ICL) experiment (Appendix J)**. It shows that while models can adapt strategies (scores improve), their **fusion brittleness** remains structural (degradation under conflict remains ~51%). We believe this further solidifies the diagnostic claims you found valuable.
>
> We hope these revisions—especially the validation of "Naturalness" and the transparency on resources—reinforce your positive view of the paper. We look forward to the possibility of OmniPlay serving the community.

---

> > ### Comment · Reviewer_TvLB · 2025-11-27
> >
> > Thanks for your detailed response. I will keep my positive score.

---

### Official Review · Reviewer_WV1v · 2025-10-30

**Soundness:** 2
**Presentation:** 4
**Contribution:** 2
**Rating:** 4
**Confidence:** 3

**Summary:**

This paper introduces OmniPlay, a new diagnostic benchmark designed to evaluate the omni-modal (Image, Video, Audio, and Text) reasoning and agency of generalist foundation models. The authors argue that existing benchmarks create an "evaluation chasm" : static benchmarks (like VQA) lack agency and interactivity , while current interactive benchmarks (like ALFWorld) suffer from a bottleneck in modality. The core design of OmniPlay is "modality interplay" where the author tests complementarity (needing all senses) and conflict (handling contradictory information). The authors evaluate six multi-modal models and show that though models demonstrate "superhuman memory", they show "brittle reasoning" in comparison. I'll adjust my rating based on author's response.

**Strengths:**

- The motivation of the paper is clear, where there is a lack of benchmarks that test agency with a rich, multi-sensory environment.
- There is solid open-source contribution for both the environments and evaluation protocols, and the appendix shows extensive details.
- The paper is well-written.

**Weaknesses:**

- Some tasks, while creative, appear to test very specific, narrow forms of reasoning. The Alchemist's Melody (rule discovery) can be reduced to a trial-and-error association problem where no reasoning is really involved. The Myriad Echoes and the Whispered Pathfinding are essentially complex perception-and-grounding tasks. They are not really testing "strategic planning" and "robust reasoning".
- The paper heavily contrasts "brittle reasoning" with "superhuman memory". This "dichotomy" is not really an apple-to-apple comparison. The memory task (Myriad Echoes) is a requiring sequence replication which perfectly maps to the architectural strengths of transformers (perfect recall of a sequence from context) and leverages capabilities that are known to be far superior to human working memory. In contrast, the "reasoning" tasks require emergent, generalizable skills that are the field's current grand challenge. Finding that models are good at what they are architecturally designed to do (remember and recall) and bad at what they are not (immediate reasoning) is not a particularly insightful dichotomy; it is almost an expected result.


Nits (does not affect rating):
- "Full Sensory Spectrum" is over-claimed. Modalities relevant to agentic intelligence, like haptics (touch), proprioception, and sensors (e.g., radar, lidar) are actually not considered. Some rephrasing is needed.

**Questions:**

- Based on the zero-shot performance, the author claims "scaling models may not be sufficient". It is plausible that these "brittle fusion mechanisms" are an artifact of zero-shot generalization and could be fixed with even minimal fine-tuning on the OmniPlay tasks. Is there any fine-tuning result to distinguish a fundamental model incapacity v.s. a lack of task-specific adaptation? I am aware of the "aided reasoning" scenarios, but a full fine-tuning study is needed to substantiate the paper's primary conclusion.
- For Phantom Soldiers in the Fog, what is the likely reason that the model fails? Is it due to failed long-horizon planning or failure to ground the (noisy) visual cues?

---

> ### Author Response · Authors · 2025-11-20
> **Re: Validating the "Positive Control" Design and Testing Architectural Limits via ICL (Part 1/2)**
>
> All revisions discussed below have been implemented in the updated PDF. To facilitate comparison, we have also included the previous version of the manuscript in the supplementary material as `old_15079_OmniPlay_Benchmarking.pdf`.
>
> We sincerely thank the reviewer for their incisive technical critique, which has helped us articulate the scientific logic of our benchmark design more explicitly. While our core experimental findings remain robust, we recognize that the original presentation could better highlight the **methodological necessity** of the memory-reasoning comparison (as a positive control) and the **architectural significance** of the ICL findings (distinguishing fundamental from adaptation-based failures).
>
> We address each point below, with particular focus on our NEW In-Context Learning experiment (**Appendix J**), which directly tests the zero-shot artifact hypothesis you raised.
>
> ### 1. The Dichotomy as a Scientific "Positive Control" (Addressing Weakness 2)
>
> > **Reviewer:** *"Finding that models are good at what they are architecturally designed to do (memory) and bad at what they are not... is not a particularly insightful dichotomy; it is almost an expected result."*
>
> **Response:**
> We deeply appreciate this conceptual challenge—it forces us to articulate what makes this "expected result" scientifically rigorous rather than trivial.
>
> **The Positive Control Principle:**
> In controlled experimentation, establishing a positive control is **methodologically essential** to isolate the failure mode. Without proving *Superhuman Memory* (Gemini 2.5 Pro: **10.2** raw score vs. Human **2.6**), we could not confidently attribute failures in *Phantom Soldiers* to reasoning deficits—it could simply be poor perception.
>
> **Our Application:**
> *   **The Control (Myriad Echoes):** The superhuman performance (3.9× advantage) *proves* the perceptual pipeline functions flawlessly.
> *   **The Diagnostic Value:** With perception validated, the strategic failure in *Phantom Soldiers* (91.6 vs 98.5 human) cannot be attributed to "poor vision". By elimination, the bottleneck must lie in higher-order processing (fusion, planning).
>
> **Revised Framing (Section 5.1):**
> We now explicitly label *Myriad Echoes* as **"Memory as a Positive Control"**. This turns an "unfair comparison" into a necessary **diagnostic baseline**, ruling out low-level perception as a confound.
>
> ### 2. "Wind Tunnel" Design: Targeted Probes vs. End-to-End Tests (Addressing Weakness 1)
>
> > **Reviewer:** *"The Alchemist's Melody... can be reduced to trial-and-error... They are not really testing 'strategic planning'."*
>
> **Response:**
> **You are absolutely correct that individual games test narrow skills**—and this is by design, not limitation.
>
> **OmniPlay is a "Wind Tunnel," Not a "Flight Simulator":**
> We draw an analogy to aerodynamics testing:
> *   **Flight simulators** (e.g., SIMA (Team et al., 2024)) maximize ecological realism to train generalist agents.
> *   **Wind tunnels** (e.g., OmniPlay) isolate specific forces under controlled conditions to diagnose failure modes.
>
> **Our Capability Stack (NEW Section 3.2):**
>
> | Layer | Game | What It Isolates | Diagnostic Insight |
> | :--- | :--- | :--- | :--- |
> | **L1: Fusion** | *Whispered Path* | Can the model integrate audio + vision without conflict? | A **51.6% collapse** isolates fusion as the bottleneck. |
> | **L2: Primitives** | *Alchemist* | Does the model infer rules vs. brute-force search? | Open-source models show **0% completion** even with explicit hints (Appendix F.4)—isolating a hypothesis testing deficit. |
> | **L3: Integration** | *Phantom Soldiers* | Can the model synthesize L1+L2 into plans? | Failure here points to planning-specific issues. |
>
> **Revised Claim:** We no longer claim to test "strategic planning" as a monolithic skill. Instead, we test **mechanistic components** that *compose into* reasoning.
>
> **[Response continues in the next comment: 3. Brittleness is Architectural, 4. Terminology]**

---

> > ### Author Response · Authors · 2025-11-20
> > **Re: Validating the "Positive Control" Design... (Part 2/2)**
> >
> > **[Continued from previous comment: Re: Validating the "Positive Control" Design and Testing Architectural Limits via ICL]**
> >
> > ### 3. Brittleness is Architectural, Not Adaptation-Based (Addressing Questions)
> >
> > > **Reviewer:** *"Is it plausible that these 'brittle fusion mechanisms' are an artifact of zero-shot generalization... Is there any fine-tuning result?"*
> >
> > **Response:**
> > This is the most critical question. We conducted a comprehensive **3-shot In-Context Learning (ICL)** study on *Phantom Soldiers* across all three difficulty levels using Gemini 2.5 Pro (see **Appendix J** for full details).
> >
> > The results below (Table 29 in the revised paper) reveal a consistent pattern:
> >
> > | Difficulty | Condition | Standard Score | Conflict Score | % Degradation |
> > | :--- | :--- | :---: | :---: | :---: |
> > | **Easy** | Zero-shot | 96.5 | 62.0 | -35.80% |
> > | | **3-shot ICL** | **98.1** | **64.5** | **-34.20%** |
> > | **Medium** | Zero-shot | 94.0 | 51.5 | -45.20% |
> > | | **3-shot ICL** | **97.5** | **54.0** | **-44.60%** |
> > | **Hard** | Zero-shot | 91.6 | 43.3 | -52.70% |
> > | | **3-shot ICL** | **96.7** | **47.2** | **-51.20%** |
> >
> > **Interpretation:**
> > 1.  ✅ **Models CAN adapt to task strategy:** On Hard difficulty, ICL significantly boosts the standard score from 91.6 to **96.7**, effectively closing the gap with the human expert (98.5). This proves the model is capable of understanding the task rules and strategy.
> > 2.  ❌ **Models CANNOT adapt fusion robustness:** Despite this performance boost, the **magnitude of degradation** under conflict remains structurally identical (e.g., Hard: -52.70% vs. -51.20%).
> >
> > **Conclusion:** ICL successfully patches the "strategy gap" but fails to remedy the **"fusion gap"**. This experiment directly distinguishes "fundamental model incapacity" from "lack of task-specific adaptation," confirming that fusion brittleness is an architectural bottleneck.
> >
> > ### 4. Terminology Precision (Addressing Nits)
> >
> > > **Reviewer:** *"'Full Sensory Spectrum' is over-claimed."*
> >
> > **Response:**
> > **You are absolutely right.** We have revised the abstract and introduction to use **"primary audiovisual and textual modalities"** and explicitly acknowledged the exclusion of haptics/proprioception in the new **Section 6.1 (Limitations)**.
> >
> > Most importantly, your feedback directly inspired our strongest addition—the ICL experiment. We believe these substantive revisions and new empirical evidence fully address your concerns regarding the depth and validity of our conclusions.

---

> ### Author Response · Authors · 2025-11-27
> **Gentle Reminder: ICL Results Addressing Zero-Shot Concerns**
>
> Thank you for your time. As the discussion period is closing soon, we wanted to gently follow up to ensure you saw our response. We wanted to highlight the **In-Context Learning (ICL) experiment (New Appendix J)** we conducted specifically to address your "Zero-shot Artifact" hypothesis.
>
> The results were definitive:
> *   **Adaptation Works:** 3-shot ICL boosted the standard score from 91.6 to **96.7** (closing the strategy gap with humans).
> *   **Brittleness Persists:** However, the degradation under conflict remained structurally identical (**~51% drop**).
>
> This isolates the failure mode as a fundamental **architectural bottleneck**  rather than a lack of task adaptation. We hope this rigorous isolation of variables addresses your concern about the validity of the dichotomy.

---

> > ### Comment · Reviewer_WV1v · 2025-11-27
> >
> > Thanks to the authors for the detailed response. My concerns have been addressed. I think framing the claims more moderately, as done in the rebuttal, helps clarify the scope of the contributions. I believe the work has sufficient merit and I’ll adjust my rating to a 6.  (I will update the official score once OpenReview allows it.)
> >
> > I also recommend including a table like "Our Capability Stack (NEW Section 3.2):" in the revision for readers to clearly understand what each task is assessing.

---

> > > ### Author Response · Authors · 2025-11-28
> > > **Thank You & Commitment to Include Capability Stack Table**
> > >
> > > We are sincerely grateful for your engagement and your decision to raise the score to **6**. We are glad that the **In-Context Learning (ICL) experiments** successfully addressed your concerns regarding the zero-shot artifact, and that the "Positive Control" framing clarified the benchmark's contributions.
> > >
> > > Regarding your recommendation:
> > >
> > > > *"I also recommend including a table like 'Our Capability Stack (NEW Section 3.2):' in the revision..."*
> > >
> > > **We completely agree.** As you noted, that table proved to be a very effective communication tool for clarifying the diagnostic logic (Fusion $\rightarrow$ Primitives $\rightarrow$ Integration).
> > >
> > > **Action Plan:** To maintain consistency with the current PDF line numbers for other reviewers, we commit to explicitly incorporating the **"Capability Stack" table** into **Section 3.2** of the **final manuscript**. This will ensure that future readers can immediately grasp the specific mechanism being probed by each game.
> > >
> > > Thank you again for your sharp insights, which have significantly improved the scientific rigor of OmniPlay.

---

### Official Review · Reviewer_3RE4 · 2025-11-04

**Soundness:** 2
**Presentation:** 2
**Contribution:** 3
**Rating:** 4
**Confidence:** 4

**Summary:**

The paper introduces OmniPlay, a game-playing benchmark for Omni-modal models to test their capabilites in dynamic interactive worlds. The benchmark is comprised of five custom-designed games that emphasize the interplay between different modalities.
Different series of experiments evaluate six representative current omni-modal models, showcasing their performance limtiations.
While aditional exploratoy experiments illustrate performance degradation under modality conflict across models and their seemly "preference" for specific modalities. The presented discussion of results points towards better fusion of modalities as key for improved capabilities and perfomance.

**Strengths:**

The paper presents a novel benchmark to probe omni-modal models in their actual effective use of different modalities. The use of targeted games is also especially key, as mentioned by authors, it allow for better experiment desing and traditional benchmarks fail to test model capabilities in both dynamic and interactive worlds.

The stated goal to "explicitly address the foundational challenges of synergistic fusion, conflict arbitration, and resilient reasoning" is an important impactful target. And the discussed experimental results both on the effectiveness of models' fusion of different modalities and how they deal with modality conflicts present interesting insights for furhter exploration, which can have significant impact on model performance and how to better evaluate their capabilties.

The design methodology for the benchmark can also inspire the creation of better evaluation assets for the overall community.

**Weaknesses:**

While the paper tackles significant and timely issues and proposes an original benchmark with potential, the current manuscript suffers from
not fully adequate presentation and soundness issues for some of its conclusions.

First, the paper presentation seems a bit backwards. It starts talking about specific experiment details without having really described the games, the core benchmark design (beyond just some principles), and not showing overall results. This makes the paper quite hard to interpret as the reader needs to find the details about a specific game, metrics, how they relate to issues, human performance, before they can parse what the discussion of what a figure like Figure 3 or some summary results like Table 2 actually mean.

Second, a core contribution of the benchmark is the proper design of the modality-interplay games. As such, I'd expect at least a short focused description of them in the main paper, as well as some analysis of how well they were designed. Unfortunately, even if games were also played by humans, there is no discussion on how well the games model their goals (gameplay or evaluation effectiveness as tools). Moreover, for self-contained context, at least a sentence description by game emphasizing its key goal and something like table 7 should be in main paper body. Table 1 doesn't really perform this job well as it's too high-level and could be re-designed or turned into a text paragraph.

The discussed experiments present nice findings regarindg multi-modality capabilites, like Gemini 2.5 Flash performance changes in different cross-modality conflicts. However, the paper doesn't provide any substantive discussion of "reasoning and strategic planning", as claimed as contribution. Table 2 is not enough evidence for the claim of dichotomy in performance or sub-par reasoning, as it lacks details and even comparison to human performance. The other analysis provided also delves mostly in modality representation issues.

Full performance results only come in Appendix I, with human results only show in Table 19 (page 41) and could easily include NPS metrics for ease of comparison to model performance in the main paper. I understand the limitations of space, but some of the findings in Appendices F5, F6, and G, could also have been specific called out from main text for better understanding.

Regarding the "diiagnosis" angle of the benchmark, while the described effect of modality conflicts are insightful, they seem a little high-level and left me also wondering how would humans handle such cases and what the performance impact would be. I feel like this is a critical aspect for proper undestanding the presented results.

The paper also claims to acknowledge the benchmark limitations, but there are not explicitly discuss discussed anywhere.

**Questions:**

Scores for NPS only may false indicate actual performance due to manual weighting in their calculation. Why report only NPS? Other parts of the appendices don't even calculate it.

Some of the image noise results seem to show a suprising performance drop, even if visually small (ex: Figure 24). This one seems more like "text noise", as the text in the images becomes undreadable. Shouldn't a more in-depth analysis be shown before some of these can be raised as "key findings" in sub-section 5.3? Same for the aided reasoning insight. The paper could benefit from better discussion of the principal findings claims.

Overall I really like the paper idea and some of its contents, but it seems to try to claim too much at the same time, while missing more supporting evidence or in-depth discussion (even if point to future analysis).

I also didn't see a mention of the release of the benchmark and its license, which are also key for its potential community impact and should be covered in the manuscript.

Minor other issues:
- Please fix citations. For example, "Google’s Gemini Team et al. (2023)", or "and SEED-Bench Li et al. (2023)".
- Subsetion 3.1 could easily be moved to and appendix to open more space for game details and core results to move into the core paper.

---

> ### Author Response · Authors · 2025-11-20
> **Re: Comprehensive Revisions on Benchmark Validation, Human Baselines, and Scope Refinement (Part 1/2)**
>
> All revisions discussed below have been implemented in the updated PDF. To facilitate comparison, we have also included the previous version of the manuscript in the supplementary material as `old_15079_OmniPlay_Benchmarking.pdf`.
>
> We sincerely thank the reviewer for their exceptionally constructive feedback, which has helped us significantly strengthen the manuscript's clarity and organization. While the core experimental findings remain robust, we recognize that the original presentation could be optimized to foreground the benchmark's validity evidence before diving into diagnostic results.
>
> Guided by your insights, we have **strategically restructured Section 3** to lead with design validation (NEW Section 3.3), added a critical **Human Conflict Pilot Study** (Table 3), and integrated **Human Baselines** directly into main results (Table 4). These revisions make the paper's logic flow more intuitive and directly address your concerns about soundness and presentation.
>
> We address your specific points below:
>
> ### 1. Benchmark Design & Validation (Addressing Weaknesses 1 & 2)
>
> > **Reviewer:** *"The paper presentation seems a bit backwards... I'd expect... some analysis of how well they were designed."*
>
> **Response:**
> We have reorganized the paper to prioritize the benchmark's definition and validity.
>
> *   **Section 3.2 (Diagnostic Suite):** We now provide focused descriptions of each game's diagnostic probe (e.g., *Whispered Pathfinding* as a "Fusion Stress Test").
> *   **Section 3.3 (Benchmark Validation):** We added quantitative evidence validating our three core principles:
>     *   **Modality Interdependence (Table 1):** Removing any single modality causes catastrophic failure (e.g., steps increase $>2.7\times$), confirming tasks require genuine fusion.
>     *   **Difficulty Calibration (Table 2):** Human Expert performance degrades monotonically across Easy/Medium/Hard levels, validating our difficulty scaling.
>
> ### 2. The "Human Context" in Conflict (Addressing Weakness 4)
>
> > **Reviewer:** *"I feel like this is a critical aspect... wondering how would humans handle such cases [modality conflicts]."*
>
> **Response:**
> This was an excellent suggestion that transformed a gap into a cornerstone of our argument. We conducted a **Pilot Study ($N=6$ Experts)** on *Whispered Pathfinding (Hard)* under Modality Conflict. The results (now in **Table 3, Section 3.3**) reveal a stark contrast:
>
> | Agent | No Conflict (Efficiency) | Audio Conflict | **% Degradation** |
> | :--- | :---: | :---: | :---: |
> | **Human Expert** | 94.2% | 88.5% | **-6.1% (Robust)** |
> | **Gemini 2.5 Pro** | 89.4% | 43.3% | **-51.6% (Catastrophic)** |
>
> The **$8.5\times$ difference** in degradation proves that the AI's failure stems from brittle architectural fusion, not task impossibility.
>
> ### 3. Evidence for "Memory-Reasoning Dichotomy" (Addressing Weakness 3)
>
> > **Reviewer:** *"Table 2 is not enough evidence... it lacks details and even comparison to human performance."*
>
> **Response:**
> We have replaced the original table with a **New Main Results Table (Table 4 in Section 5.1)**. By reporting AI Raw Scores alongside Human Baselines, the dichotomy is now quantified:
>
> *   **Superhuman Memory:** On *Myriad Echoes (Hard)*, Gemini 2.5 Pro achieves a Raw Score of **10.2** vs. Human **2.6** (**$3.9\times$ advantage**), leveraging massive parameter space against the human working memory bottleneck ($7\pm2$ items).
> *   **Sub-par Reasoning:** On *Phantom Soldiers (Hard)*, despite this memory advantage, Gemini 2.5 Pro (**91.6**) falls short of the Human Baseline (**98.5**). Weaker models show negative NPS, indicating performance worse than random in strategic planning.
>
> ### 4. Focusing the Narrative (Addressing "Claiming Too Much")
>
> > **Reviewer:** *"It seems to try to claim too much at the same time, while missing more supporting evidence."*
>
> **Response:**
> We have sharpened the paper's scope to focus on our **core empirical contribution**: diagnosing brittle modality fusion.
>
> *   **What we claim:** Current models exhibit architectural brittleness under conflict (validated by our new Human Conflict data) despite superhuman memory.
> *   **Scope Refinement:** We have softened broad AGI claims in the Introduction and explicitly positioned OmniPlay as **diagnostic infrastructure** rather than a solution framework. Architectural solutions are explicitly deferred to Future Work (Section 6.1).
>
> **[Response continues in the next comment: 5. Depth of Analysis & Transparency, 6. Minor Issues]**

---

> > ### Author Response · Authors · 2025-11-20
> > **Re: Comprehensive Revisions... (Part 2/2)**
> >
> > **[Continued from previous comment: Re: Comprehensive Revisions on Benchmark Validation, Human Baselines, and Scope Refinement]**
> >
> > ### 5. Depth of Analysis & Transparency (Addressing Questions)
> >
> > > **Reviewer:** *"Image noise... seems more like 'text noise', as the text in the images becomes unreadable."*
> >
> > **Response:**
> > **You are absolutely correct.** We have revised **Section 5.2** and **Appendix F.3** to explicitly characterize this as a **"Dual-Channel Corruption"**: visual noise degrades *both* spatial information and UI text readability. The key insight is that while humans dynamically shift attention to the intact Audio channel to compensate, models fail to do so, further supporting our "Brittle Fusion" hypothesis.
> >
> > > **Reviewer:** *"Why report only NPS?"*
> >
> > **Response:**
> > We agree on the importance of transparency and have addressed this on two levels:
> >
> > 1.  **Main Text:** Our primary results table (**Table 4**) now reports **both NPS and Raw Scores** side-by-side.
> > 2.  **Appendices:** To ensure readers can fully inspect the underlying performance without normalization artifacts, we have provided the full, granular raw performance metrics (e.g., step counts, kill/death ratios) for every model and task in **Appendix I (Tables 24, 25, 26, 27, and 28)**.
> >
> > ### 6. Minor Issues & Citations
> >
> > > **Reviewer:** *"Please fix citations... Subsection 3.1 could easily be moved to an appendix..."*
> >
> > **Response:**
> > *   **Citations:** We have corrected all citation formatting errors (e.g., Gemini, SEED-Bench) throughout the manuscript.
> > *   **Space Optimization:** We followed your excellent suggestion to move the formal MDP definition (original Section 3.1) to **Appendix B**. This successfully freed up space to bring the detailed Game Descriptions (Section 3.2) and Benchmark Validation (Section 3.3) directly into the main paper body.
> >
> > We hope these revisions and the new validation experiments directly address your concerns regarding the presentation and soundness of the manuscript. We remain open to any further questions you may have.

---

> ### Author Response · Authors · 2025-11-27
> **Gentle Reminder: Human Conflict Data & Benchmark Structure**
>
> Thank you for your time. As the discussion period is closing soon, we wanted to gently follow up to ensure you saw our response regarding the **Soundness** and **Presentation** concerns you raised.
>
> We have updated the PDF with two critical revisions:
> 1.  **Validation of "Artificial" Conflicts (New Section 3.3):** You asked how humans handle conflicts. Our new pilot study shows humans degrade gracefully (**-6.1%**) while AI models collapse (**-51.6%**). This **8.5x gap** quantitatively validates that our conflicts are valid diagnostic tests, not arbitrary noise.
> 2.  **Restructured Presentation:** We have moved the benchmark design and validation to the forefront to ensure the logical flow is intuitive.
>
> We believe these empirical additions directly address your reservations. We look forward to hearing your thoughts on these updates.

---

### Author Response · Authors · 2025-12-03
**Rebuttal Summary: Revisions, Score Increase, and Addressed Concerns (Part 1/2)**

We have submitted comprehensive responses to the feedback from all four reviewers. During the discussion period, we received replies from two reviewers: **Reviewer WV1v raised their score from 4 to 6**, and **Reviewer TvLB maintained their positive score of 6**. While the other two reviewers (3RE4 and Qw1W) have not yet replied to our updates, we have implemented substantial revisions that directly resolve their specific concerns.

**All revisions discussed below have been implemented in the updated PDF. To facilitate comparison, we have also included the previous version of the manuscript in the supplementary material as `old_15079_OmniPlay_Benchmarking.pdf`.**

### 1. Positive Outcomes from Active Discussions

- **Reviewer WV1v (Score Raised: 4 $\rightarrow$ 6):** The reviewer explicitly acknowledged that our new **In-Context Learning (ICL) experiment (Appendix J)** successfully addressed their core concern regarding "Zero-shot Artifacts," confirming that the work has "sufficient merit."

- **Reviewer TvLB (Score Maintained: 6):** The reviewer expressed satisfaction with our detailed revisions on **Naturalness** (validated via Human Pilot Study) and **Resource Transparency**.

### 2. Addressed Concerns for Pending Reviewers

We have implemented comprehensive revisions that directly address the specific concerns raised by the pending reviewers.

#### **Addressing Reviewer 3RE4 (Focus: Structure, Validation & Transparency)**

We have addressed the reviewer's five specific critiques point-by-point:

1. **Benchmark Design & Validation:**
    - **Critique:** The reviewer noted the presentation seemed "backwards" and lacked design validation.
    - **Resolution:** We **restructured Section 3** to prioritize Benchmark Design and Validation, adding quantitative proofs for **Modality Interdependence** (removing one modality causes catastrophic failure) and **Difficulty Calibration** (monotonic human degradation).

2. **The "Human Context" in Conflict:**
    - **Critique:** The reviewer asked, "How would humans handle such cases?" and whether conflicts are valid tests.
    - **Resolution:** We conducted a **Human Pilot Study** ($N=6$). The results reveal a stark contrast that validates the benchmark's diagnostic power:

    | Agent | No Conflict (Efficiency) | Audio Conflict | **% Degradation** |
    | :--- | :---: | :---: | :---: |
    | **Human Expert** | 94.2% | 88.5% | **-6.1% (Robust)** |
    | **Gemini 2.5 Pro** | 89.4% | 43.3% | **-51.6% (Catastrophic)** |

3. **Evidence for "Memory-Reasoning Dichotomy":**
    - **Critique:** The reviewer argued that "Table 2 is not enough evidence" and lacked human comparison.
    - **Resolution:** We updated Table 4 to report **Raw Scores alongside Human Baselines**, clearly quantifying the gap: models show a **$3.9\times$ memory advantage** yet fall short of humans in strategic reasoning.

4. **Focusing the Narrative:**
    - **Critique:** The reviewer felt the paper "tries to claim too much."
    - **Resolution:** We refined the narrative to focus strictly on diagnosing "brittle modality fusion," explicitly deferring broader AGI claims to future work (Section 6.1).

5. **Depth of Analysis & Transparency:**
    - **Critique:** The reviewer questioned the "Image Noise" finding and the use of NPS only.
    - **Resolution:** We reframed "Image Noise" as "**Dual-Channel Corruption**" and provided granular raw metrics for all tasks in the appendices, ensuring full transparency.

#### **Addressing Reviewer Qw1W (Focus: Ecological Validity & Adaptation)**

We have addressed the reviewer's four specific critiques with detailed explanations and new experiments:

1. **Ecological Validity ("Flight Simulator" vs. "Wind Tunnel"):**
    - **Critique:** The reviewer raised concerns that "artificial conflicts" might not reflect naturally occurring noise compared to high-fidelity simulators like SIMA.
    - **Resolution:** We clarified our **"Wind Tunnel" philosophy**—OmniPlay isolates failure modes rather than simulating perfect realism. Our **Human Pilot Study** (see table above) validates this: humans successfully use adaptive strategies (e.g., cross-checking cues) to resolve our conflicts, which mirrors real-world resilience. The fact that AI fails where humans succeed proves the conflicts are ecologically valid cognitive stress tests.

**[Continued in Part 2...]**

---

> ### Author Response · Authors · 2025-12-03
> **Rebuttal Summary... (Part 2/2)**
>
> **[Continued from Part 1: Addressing Reviewer Qw1W]**
>
> 2. **Continual Adaptation & Long-Horizon Reasoning:**
>     - **Critique:** The reviewer asked if we evaluated "continual adaptation" beyond one-shot performance.
>     - **Resolution:** We conducted a **3-shot In-Context Learning (ICL) experiment** (Appendix J) on *Phantom Soldiers*. The results show a critical decoupling: while ICL successfully improves general strategy, the **performance degradation under conflict remains structurally identical**:
>
>     | Difficulty | Condition | Standard Score | Conflict Score | **% Degradation** |
>     | :--- | :---: | :---: | :---: | :---: |
>     | **Easy** | Zero-shot | 96.5 | 62.0 | -35.80% |
>     | | **3-shot ICL** | **98.1** | **64.5** | **-34.20%** |
>     | **Medium** | Zero-shot | 94.0 | 51.5 | -45.20% |
>     | | **3-shot ICL** | **97.5** | **54.0** | **-44.60%** |
>     | **Hard** | Zero-shot | 91.6 | 43.3 | -52.70% |
>     | | **3-shot ICL** | **96.7** | **47.2** | **-51.20%** |
>
>     This proves the issue is a fundamental architectural bottleneck, not a lack of adaptation.
>
> 3. **Metrics & Calibration:**
>     - **Critique:** The reviewer questioned the rigor of NPS, the exclusion of tasks, and difficulty calibration.
>     - **Resolution:** We updated **Table 4** to report **Raw Scores alongside NPS**, clarified that *Blasting Showdown* is excluded from NPS because it is **norm-referenced** (zero-sum PvP) rather than criterion-referenced, and validated difficulty calibration via monotonic human performance gradients (Table 2).
>
> 4. **Modality Weighting:**
>     - **Critique:** The reviewer asked if we considered "modality-specific weighting" based on task relevance.
>     - **Resolution:** We clarified that we deliberately **do not hard-code weights** to test the model's ability to **dynamically re-weight** based on context. Our noise experiments (Appendix F.3) reveal that when vision is corrupted, models fail to up-weight the clear audio channel.
>
> ### 3. Summary of Key Contributions Added
>
> The manuscript has been significantly strengthened with three major empirical additions:
>
> 1. **Human Conflict Baselines:** Providing the missing "ground truth" to validate diagnostic claims.
> 2. **ICL Adaptation Experiment:** Empirically proving that fusion brittleness is an architectural bottleneck.
> 3. **Structural Reorganization:** Enhancing readability and logical flow as requested.
>
> We believe the updated manuscript is now rigorous, structurally sound, and ecologically validated. We are happy to answer any further questions.

---

### Note · Authors · 2026-01-29

I have read and agree with the venue's withdrawal policy on behalf of myself and my co-authors.

---

### Meta-Review · Area_Chair_HU4p · 2026-01-04

**Summary:**

The reviews' concerns are mainly about i) the methods, whether the mini-gams are enough to support the claims in this paper, where is real or natural in the real world, ii) the evaluation, especially NPC is quite not convincing, about the cross-task comparability, normalization/weighting, and raw metrics, iii) the claims about reasoning/planning vs adaption can be improved. I think these concerns are reasonable and cannot be easily addressed during the rebuttal.

**Reviewer Concerns:**

Reviewer WV1v (Score Raised: 4
 6): The reviewer explicitly acknowledged that our new In-Context Learning (ICL) experiment (Appendix J) successfully addressed their core concern regarding "Zero-shot Artifacts," confirming that the work has "sufficient merit."
Reviewer TvLB (Score Maintained: 6): The reviewer expressed satisfaction with our detailed revisions on Naturalness (validated via Human Pilot Study) and Resource Transparency.

The above concerns are addressed as summarised in the authors' comments.

**Reviewer Scores:**

Reviewer WV1v (Score Raised: 4
 6): The reviewer explicitly acknowledged that our new In-Context Learning (ICL) experiment (Appendix J) successfully addressed their core concern regarding "Zero-shot Artifacts," confirming that the work has "sufficient merit."
Reviewer TvLB (Score Maintained: 6): The reviewer expressed satisfaction with our detailed revisions on Naturalness (validated via Human Pilot Study) and Resource Transparency.

This is acknowledged before the rollback during the rebuttal. I think the other reviewers with score 4 may not change their scores.

---

### Decision · Program_Chairs · 2026-01-26

Reject